# ALAN is a computational approach that interprets genomic findings in the context of tumor ecosystems

Hannah E. Bergom[1,2,10], Ashraf Shabaneh[1,3,10], Abderrahman Day [1,2,3,10], Atef Ali[1,2], Ella Boytim[1,2], Sydney Tape[1,2], John R. Lozada[1], Xiaolei Shi[1], Carlos Perez Kerkvliet[1], Sean McSweeney[1], Samuel P. Pitzen [4,5], Megan Ludwig[6], Emmanuel S. Antonarakis[1,2,4], Justin M. Drake[1,6,7], Scott M. Dehm [4,7,8], Charles J. Ryan[1,2,4,9], Jinhua Wang[1,3,4] & Justin Hwang [1,2,4✉]

Gene behavior is governed by activity of other genes in an ecosystem as well as context-specific cues including cell type, microenvironment, and prior exposure to therapy. Here, we developed the Algorithm for Linking Activity Networks (ALAN) to compare gene behavior purely based on patient -omic data. The types of gene behaviors identifiable by ALAN include co-regulators of a signaling pathway, protein-protein interactions, or any set of genes that function similarly. ALAN identified direct protein-protein interactions in prostate cancer (AR, HOXB13, and FOXA1). We found differential and complex ALAN networks associated with the proto-oncogene MYC as prostate tumors develop and become metastatic, between different cancer types, and within cancer subtypes. We discovered that resistant genes in prostate cancer shared an ALAN ecosystem and activated similar oncogenic signaling pathways. Altogether, ALAN represents an informatics approach for developing gene signatures, identifying gene targets, and interpreting mechanisms of progression or therapy resistance.

[1] Department of Medicine, University of Minnesota Masonic Cancer Center, Minneapolis, MN, USA. [2] Division of Hematology, Oncology and Transplantation, University of Minnesota, Minneapolis, MN, USA. [3] Institute for Health Informatics, University of Minnesota, Minneapolis, MN, USA. [4] Masonic Cancer Center, University of Minnesota, Minneapolis, MN, USA. [5] Graduate Program in Molecular, Cellular, and Developmental Biology and Genetics, University of Minnesota, Minneapolis, MN, USA. [6] Department of Pharmacology, University of Minnesota, Minneapolis, MN, USA. [7] Department of Urology, University of Minnesota, Minneapolis, MN, USA. [8] Department of Laboratory Medicine and Pathology, University of Minnesota, Minneapolis, MN, USA. [9] Prostate Cancer Foundation, Santa Monica, CA, USA. [10] These authors contributed equally: Hannah E. Bergom, Ashraf Shabaneh, Abderrahman Day. ✉email: jhwang@umn.edu

Knowledge of gene behavior has aided scientific discovery in deciphering various diseases including those of cancerous origin[1]. For instance, the development and progression of many cancers are associated with functional loss of tumor suppressor genes such as *TP53* or functional gains of oncogenes such as *MYC*[2]. Further, specific forms of gene dysregulation events are utilized as clinical predictive biomarkers or precision therapy targets in both solid and liquid tumors, including genes such as *ERBB2*[3], *EGFR*[4], and *BCR-ABL*[5]. High levels of Androgen Receptor Variant 7 (AR-V7) in prostate cancer, a splice variant of *AR* that lacks the ligand binding domain, is associated with resistance to AR-targeted therapies (ARTs)[6]. Many cancers including breast, ovarian, and prostate with mutations in BRCA1/2 are clinically actionable due to their response to PARP inhibitors[7,8]. In each of these instances, however, some patients exhibit unpredictable therapeutic responses. Further interpretation of the gene networks with consideration of disease specificity or patient response may identify genomic features that could be purposed to predict response or outcomes with greater accuracy.

Consortium efforts have accrued DNA (genomics) data based on efforts including The Cancer Genome Atlas (TCGA) and AACR GENIE[9], RNA (transcriptomics) data have been accrued based on efforts including TCGA, Stand Up 2 Cancer (SU2C)[10], and Genotype-Tissue Expression (GTEx) Project[11]. Proteomic/phosphoprotein data can be found in data portals including the in silico human Surfaceome[12] and the Human Protein Atlas[13]. Lastly, epigenomic data and metabolomic databases include Cistrome-GO[14] and the Consortium of Metabolomics Studies (COMETS)[15]. These datasets can be evaluated using various informatics solutions to resolve how gene behavior is associated with clinical phenotypes. This has led to routine use of gene panels in molecular diagnostic assays to enhance patient subtyping, outcome prediction, treatment recommendations, and other diagnostic elements to understand tumor behavior[16–18]. This includes the Prosigna panel PAM50 which is used for breast cancer subtyping[19]. While classifier approaches to patient stratification are meaningful to distinguish subtypes, they are limited because DNA-based panels don't encompass relationships between genes. Examination of gene networks using transcriptomic or even proteomic data will contextualize the complex features of gene behavior, which can be purposed to aid in treatment decisions or for the development of therapies.

Many computational approaches have been developed to understand and interrogate gene behavior based on abundance-level data from transcriptomic or proteomic approaches. Correlation approaches identify similar patterns of gene expression across patients between gene pairs. This method does not consider expression values outside of the gene pair of interest. Hierarchical clustering approaches group genes based on similarities. These outputs establish distances between two genes based on the expression values of the gene pair in relation to all other genes. Enrichment approaches such as Gene Set Enrichment Analysis (GSEA)[20], identify the enrichment of functionally related gene sets when comparing two biological states. This approach relies on static definitions of biological pathways. In certain cases, there are true differences in a signaling pathway when a gene is active in different tissues or disease states, and this activity would be better measured through nonconventional or modified gene sets. Lastly, artificial intelligence and machine learning (AI/ML) are used to study gene behavior in many ways. These algorithms often stratify genes into classes[21–23], in which genes that fall into each class behave similarly. In the case of AI/ML, the grouping of the genes may not be transparent to a biologist, and the outcomes are influenced by the user-defined architecture of the specific model. Altogether, an approach that considers the global gene expression patterns, as well as context cues, could work with all such tools and enhance understanding of gene behavior through genomic datasets.

While current informatics tools classify signaling pathways and biomarkers, understanding of gene behavior is an entirely distinct scientific objective that requires a tool that evaluates the biological activities of the same gene in multiple contexts. ALAN networks represent a mechanism to measure how one gene behaves. For a gene, the ALAN gene network encompasses information of the varying degrees of how a gene is related to all other genes based on expression data across many patient samples. To further account for differences in gene behavior governed by all other genes, cell types, microenvironments, disease stages, and the treatment status of patients, we must consider the relationships of gene networks of multiple or even all genes within ALAN ecosystems. In this study to address this limitation, we have developed an informatics tool to study gene ecosystems and all underlying gene networks within any context. We named this computational pipeline the Algorithm for Linking Activity Networks (ALAN). The Algorithm for Linking Activity Networks (ALAN) is an algorithm that identifies and compares the behavior of genes. ALAN users must first identify the relevant clinical context, the associated omic datasets, as well as the genes of interest. ALAN is then used to compare the behavior of genes of interest based on the context of the input data. Using ALAN, we identified cancer resistance mechanisms and observed changes in gene behavior in tumors that progressed towards advanced disease.

## Results

**Development of an Algorithm to Link Activity Networks (ALAN).** We developed ALAN as an algorithm to identify and compare the behavior of genes (Fig. 1). Presuming the input assumptions are met, ALAN can be used to examine patient data from many sample types, including both bulk and single-cell as well as many technologies, such as RNA sequencing or Mass Spectrometry. These outputs generally yield abundance profiles in distinct units (TPM, FPKM, RSEM, protein intensity or abundance etc.) but many public datasets have already undergone preprocessing, including log transformation or normalization. The first step of the ALAN algorithm uses a ranked-based association method to generate ALAN profiles which are contained within ALAN Output – Matrix 1. In this matrix, the correlation coefficient represents the similarity between gene expression patterns across all patients for two genes. The correlation coefficients for a single gene against every other gene represents an ALAN profile. The second step of the ALAN algorithm uses a linear-based association method to generate ALAN gene networks which are contained within ALAN Output – Matrix 2. In this matrix, the correlation coefficient represents the similarity between the behavior of one gene with respect to all other genes detected in the dataset. Genes with highly similar networks as indicated by correlation score of above 0.7 share an ALAN gene ecosystem. We thus sought to use these gene ecosystems to analyze patient data from the perspective of cancer progression and therapy resistance.

**ALAN outputs predict critical AR activity in mCRPC.** The Androgen Receptor (*AR*) is critical for the development of prostate cancer (PC) and remains a critical target in both metastatic prostate cancer (mPC) and metastatic castration-resistant prostate cancer (mCRPC). Current standard-of-care agents for mCRPC inhibit the synthesis of AR ligands or signaling of AR. While these AR-inhibiting therapies (ART) are initially effective, a subset of patients inevitably develop resistance to these therapies. *AR* mutations, amplification, or overexpression remain

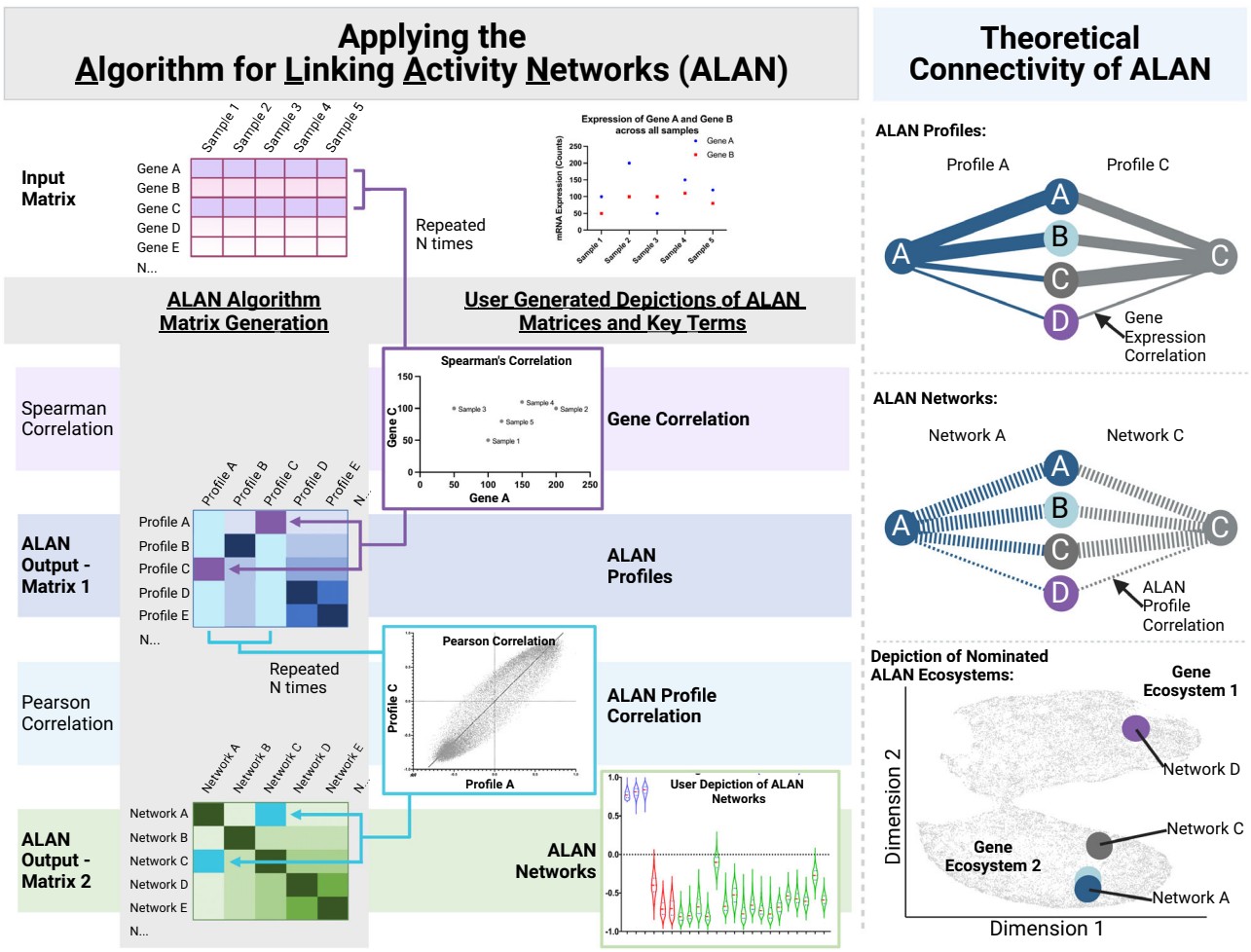

**Fig. 1 Data inputs, outputs, and their visual depictions when using the Algorithm for Linking Activity Network (ALAN).** **Applying the Algorithm for Linking Activity Networks (ALAN).** Workflow depicting the ALAN algorithm which includes input matrix, matrix generation, user-generated depictions of ALAN data matrices and key terms. **Theoretical Connectivity of ALAN.** Individual genes are depicted as circles with their respective names (A, B, C, and D). The strength of the correlation between two genes is represented by the thickness of the line. In ALAN Profiles, Profile A (blue) and Profile C (grey) depicted for genes A and C, where the strength of the correlation between the expression patterns of genes (A-A, A-B, A-C, A-D, or C-A, C-B, C-C, C-D) across all samples is represented by the thickness of the line. ALAN Profiles are derived from gene expression correlations, represented by solid lines, across all pairs of genes. In ALAN Network, Network A (blue) and Network C (grey) are depicted for genes A and C, where the strength of the correlation between ALAN profiles (A-A, A-B, A-C, A-D, or C-A, C-B, C-C, C-D) across all genes is represented by the thickness of the line. ALAN Networks are derived from ALAN Profile Correlations, represented by dashed lines, across all pairs of ALAN Profiles. In Depiction of Nominated ALAN Ecosystems, multiple ALAN networks are compared in a two-dimensional space where Networks A and C being more similar are closer together on the plot, whereas shaded dots that are not highlighted represent other ALAN networks for all detected genes.

prevalent in patients that develop resistance to multiple therapies, which reinforce that AR functions remain critical in advanced cancers[6,10,24–27]. To examine the gene networks in prostate cancer, we accrued data from 946 individuals in which whole transcriptome sequencing (WTS) was conducted on normal tissue, primary cancer, and metastatic tissue. The data were obtained through the Genotype-Tissue Expression (GTEx) project[11], The Cancer Genome Atlas (TCGA), and Stand Up to Cancer 2019 (SU2C 2019)[10]. We also confirmed such interactions in protein abundance data from primary prostate cancer patients[28].

We first tested the functionality of ALAN outputs by examining if AR would expectedly exhibit similar behavior as known co-factors. To benchmark the similarity of AR and its co-factors, we also compared these ALAN outputs to genes that are recurrently co-amplified with AR, even in primary prostate cancers. To identify co-amplified genes, in a cohort of 492 primary prostate cancer tumors, we found that AR resided in a

focal amplicon on Xq12 with 9 additional genes based on GISTIC 2.0 outputs[29,30] (Fig. 2a). In mCRPCs[10], we examined the ALAN outputs to compare gene behavior of AR, its co-factors FOXA1, HOXB13, GRHL2, NCOA2, to 8 other genes in Xq12 that were detected in the mCRPC transcriptome (Fig. 2b). This reveals that while *AR*, *HOXB13*, and *FOXA1* had similar gene behavior as defined by aggregate expression patterns across the transcriptome, this behavior was dissimilar to other genes located within the *AR* focal amplicon and suggests that they are functionally dissimilar to *AR* in mCRPC. To explore additional intrachromosomal interactions with *AR*, we categorized the chromosomal location of all 2244 genes within the ALAN *AR* network signature. Of all the genes in the ALAN *AR* network signature, only 3.8% were located on the X chromosome. However, the ALAN *AR* network signature contained the co-factors *FOXA1* (14q21) and *HOXB13* (17q21), the proto-oncogene *MYC* (8q24), and coactivator *NCOA2* (8q13) (Fig. 2c). In this regard, ALAN predicted known interactions with *AR* in mCRPC including

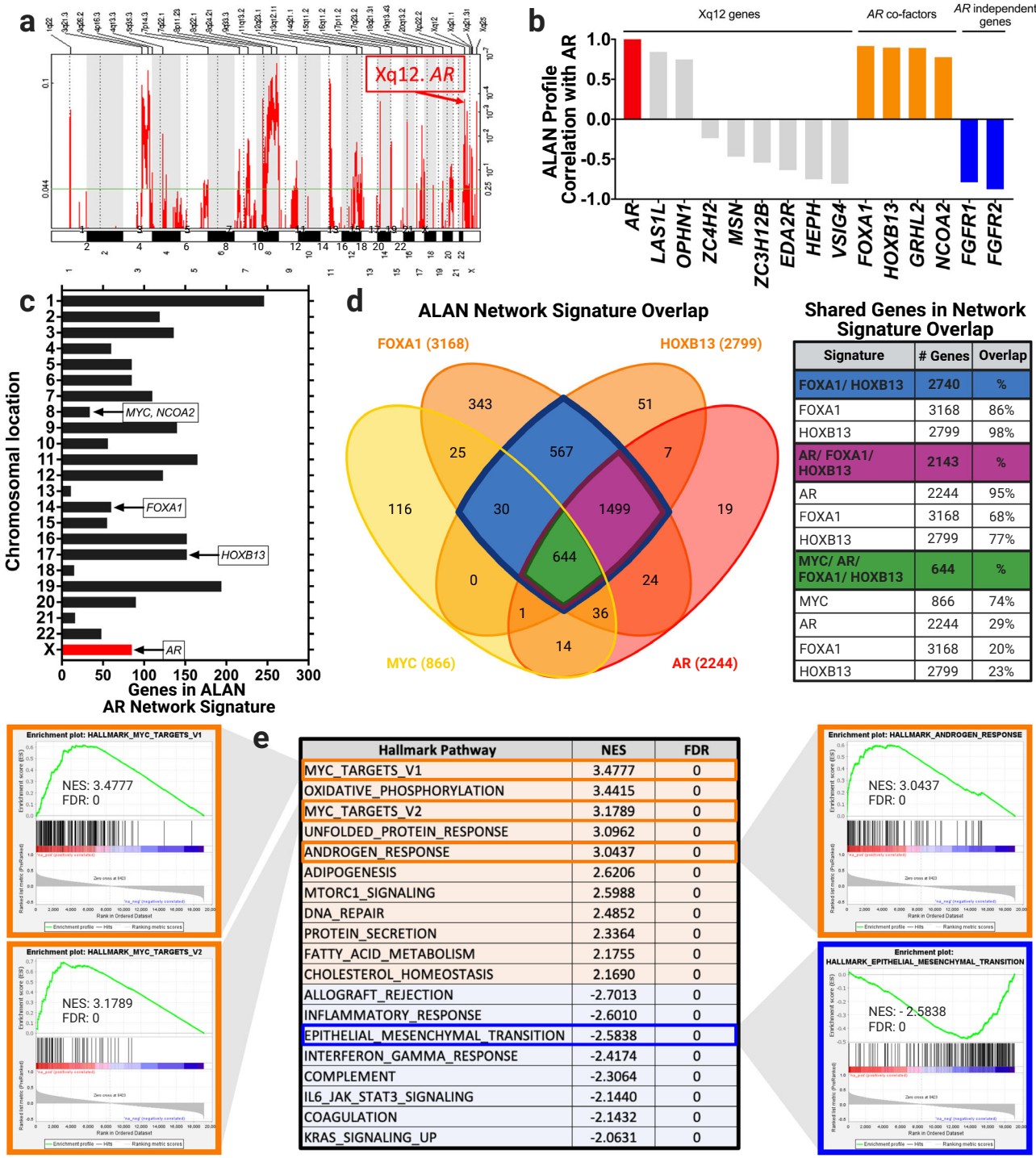

**Fig. 2 ALAN identifies similar gene behavior between AR and key prostate cancer genes that are intra-chromosomal. a** GISTIC2 analysis of primary prostate adenocarcinomas highlighting the focal amplification of AR (Xq12, $n = 9$) as a recurrent copy number gain (TCGA PRAD, $n = 492$). **b** ALAN profile correlations with AR are shown for other Xq12 genes (gray), AR co-factors (orange), and AR-independent genes (blue). **c** Chromosomal location of genes in the ALAN AR network signature. Highlighted genes have implications with AR in prostate cancer such as transcription factors MYC (8q24), FOXA1 (14q21), HOXB13 (17q21), and transcriptional co-activator NCOA2 (8q13). **d** Visual depiction of ALAN network signature overlap between AR (red), MYC (yellow), FOXA1 (orange), and HOXB13 (orange) and their percentage of shared genes. **e** GSEA Enrichment Plots with NES and FDR statistics of the AR ALAN profile and various Hallmark Gene Signatures derived from ALAN Matrix 1.

association with transcription factors HOXB13[31,32], FOXA1[31,32], and the proto-oncogene MYC[33–35]. ALAN outputs were robust compared to pairwise Spearman or Pearson's correlations (Supplementary Table 1). Our results indicate that the behaviors of genes identifiable by ALAN are powered to identify

intrachromosomal relationships between genes as a factor of gene behavior in mCRPC patients.

We further explored the networks of these genes in mCRPC by examining the similarities of their ALAN network signatures (Fig. 2d). As depicted in the Venn diagram, AR shared 95% of the

genes in its 2244 ALAN network signature with *FOXA1* and *HOXB13*, whereas 74% of the 866 genes in the *MYC* ALAN network signature were shared with *AR*, *FOXA1*, and *HOXB13*. This indicates that these genes not only have similar gene profiles (Fig. 2b), but also have similar gene networks (Fig. 2d) indicating that they reside in the same gene ecosystem in mCRPC. Our results support the hypothesis that ALAN was able to identify these previously known and essential critical gene networks in mCRPC through transcriptomic data. To support these results, we have also examined the AR and FOXA1 interactions through protein abundance data derived from prostate cancer biopsies[28]. As compared to a pairwise Spearman correlation of AR and FOXA1 protein expression levels (Supplementary Figure 1b), ALAN identified that AR and FOXA1 are in a same ALAN ecosystem (Supplementary Figure 1b). This indicated that ALAN can be utilized to compare gene behavior through protein data as well as transcriptomic data.

We also designed the ALAN outputs to be integrated directly into current genomic analytical tools, such as Gene Set Enrichment Analysis (GSEA)[20]. To evaluate the *AR* ALAN profile through conventional means, we conducted GSEA analyses on ALAN outputs from the mCRPC samples (Fig. 2e). Of the fifty hallmark gene sets from MSigDB[36], we found that, as hypothesized, the ALAN *AR* profile was associated with the Androgen response signature (NES = 3.04, FDR = 0) and two MYC Hallmark gene sets (MYC Targets V1 and V2, NES = 3.48, 3.18, FDR = 0, 0). Additionally, we found that the *AR* ALAN network was de-enriched of signatures, such as epithelial-to-mesenchymal transition (EMT, NES = −2.58, FDR = 0), which has been recently identified as an upregulated signaling pathway in mCRPC patients that have developed resistance to ART[37,38]. While the *AR* ALAN profile is enriched of pathways involving *MYC* and androgen signaling, it is de-enriched of pathways associated with metastasis and therapeutic resistance, such as EMT. This suggests that ALAN is identifying AR signaling and therapeutic resistance as potentially separate mechanisms. These observations bolster the biological findings and display both the value and ease of integrating ALAN outputs with current informatics analyses (Fig. 2E).

**ALAN mapping of MYC activity in prostate tumor progression and subtypes of breast cancer.** Previous literature has indicated that AR and MYC become co-regulators in prostate cancer, but only as a function of disease progression[33–35]. We thus expected that AR and MYC would have convergent signaling in prostate cancer and mCRPC, but to lesser degrees in normal prostate tissue. Upon analyzing AR and MYC in each clinical setting, the ALAN outputs robustly supported this biological relationship and nominated AR and MYC in the same ALAN ecosystem in mCRPC (Fig. 3a, Supplementary Table 2). Notably, pairwise Spearman's or Pearson's correlations of AR and MYC expression levels yielded limited significance when ALAN outputs were robust. These results demonstrate that ALAN outputs, which accounts for relationships of a gene with all others, reflect a cancer stage-specific functional relationship between AR and MYC. To further this analysis, we compared the ALAN gene networks of *AR*, *FOXA1*, and *MYC* using the *AR* ALAN network signature. As prostate tissue progressed from normal, to primary, to mCRPC, we observed a gradient of increasing signature scores between *MYC* and the *AR* ALAN network signature with the greatest association in mCRPC. This data indicates that while these genes share an ecosystem in mCRPC, ALAN directly visualizes these changes in networks as an evolution of gene behavior (Fig. 3b). Altogether, these ALAN analyses allowed us to observe global changes in networks and ecosystems for the same

genes across tissue with expectedly distinct histopathological behavior.

While our initial observations examined the behavior of proto-oncogene *MYC* across various stages of prostate cancer, we sought to examine the behavior of *MYC* in other cancer types, including melanoma, lung, ovarian, pancreatic, and breast. To investigate whether *MYC* exhibited distinct behavior in these additional cancer types, we utilized ALAN to examine the *MYC* network signature in each context. Transcription data was obtained from The Cancer Genome Atlas (TCGA) on samples of melanoma, lung, ovarian, and pancreatic cancer and the five PAM50 molecular subtypes of breast cancer (Basal, ER2, Luminal A, Luminal B, normal-like). Each signature was built using the top 500 genes associated with *MYC* in that cancer type or cancer subtype. Interestingly, across the four cancer types, only 3.35% of genes are shared by at least one other cancer type (Fig. 3c). In the cross-subtype comparison of breast cancers, 82.6% of all genes across the five ALAN *MYC* network signatures were unique to one subtype and only 2.8% of genes (69 total including *MYC*) were shared by at least three subtypes (Fig. 3d). Of those 69 genes, three of them, *EIF3E*, *MLLT6* and *MYC*, are indicated as cancer-causing genes by The Sanger Institute[39]. Given these distinct ALAN outputs of the *MYC* ALAN network signatures across multiple cancers and within cancer subtypes, we used GSEA to determine if the ALAN network signature was enriched of MYC activity. Despite having remarkably distinct ALAN networks, the MYC_UP.V1_UP gene signature shows high enrichment with all cancers and all 5 subtypes of breast cancer, except pancreatic (Fig. 3e). These results indicate that ALAN is able to disambiguate the genes that exhibit similar behavior as MYC across multiple cancers and within subtypes and these ALAN gene networks were enriched of Hallmark MYC activity. This highlights the utility of ALAN in finding nuance differences of an oncogene when it is active across different cancer types and cancer subtypes.

**Identification of genes and pathways that predict resistance through ALAN.** In addition to examining reported interactions, we sought to elucidate critical biology by aggregating the consistent interactive patterns of multiple ALAN gene networks, or their ALAN gene ecosystems. We particularly examined the gene ecosystems of *AR* and co-factors (*HOXB13*, *FOXA1*) as well as several genes that regulated enzalutamide resistance including *CDK6*[40], *FGFR1/2*[41], *ETV5*[42,43], *LEF1*[44], *CREB5*[45]. The other genes that share an ecosystem with these two groups of genes were identified by a correlation coefficient above 0.7 in ALAN Output – Matrix 2 across all genes in the group. Upon examining the **E**nzalutamide **R**esistance **A**LAN **E**cosystem (ERAE, genes = 1287) and the **A**R and **C**o-factors **A**LAN **E**cosystem (ACAE, genes = 2143), we observed the genes within each ecosystem co-segregated but that the two ALAN ecosystems were divergent and had 0 overlapping genes (Fig. 4a and b). To determine the significance of this result, we have conducted a hypergeometric test, using a normal approximation to the hypergeometric distribution. This result of zero overlapping genes given these two groups is highly significant (*p* value < 0.0001) indicating that this result was not due to chance. The expected number of overlapping genes based on chance is 144 given these two groups. Lastly, ALAN considered these genes to behave similarly, while pairwise Spearman or Pearson's correlations often yielded insignificant comparisons (Supplementary Table 3).

We also examined if the ERAE genes could reflect the clinical response in an individual mCRPC patient towards treatment progression (Fig. 4c). We examined scRNA seq data obtained from paired biopsy samples from one patient pre-enzalutamide treatment and post-therapy resistance[46]. Due to the sparse nature

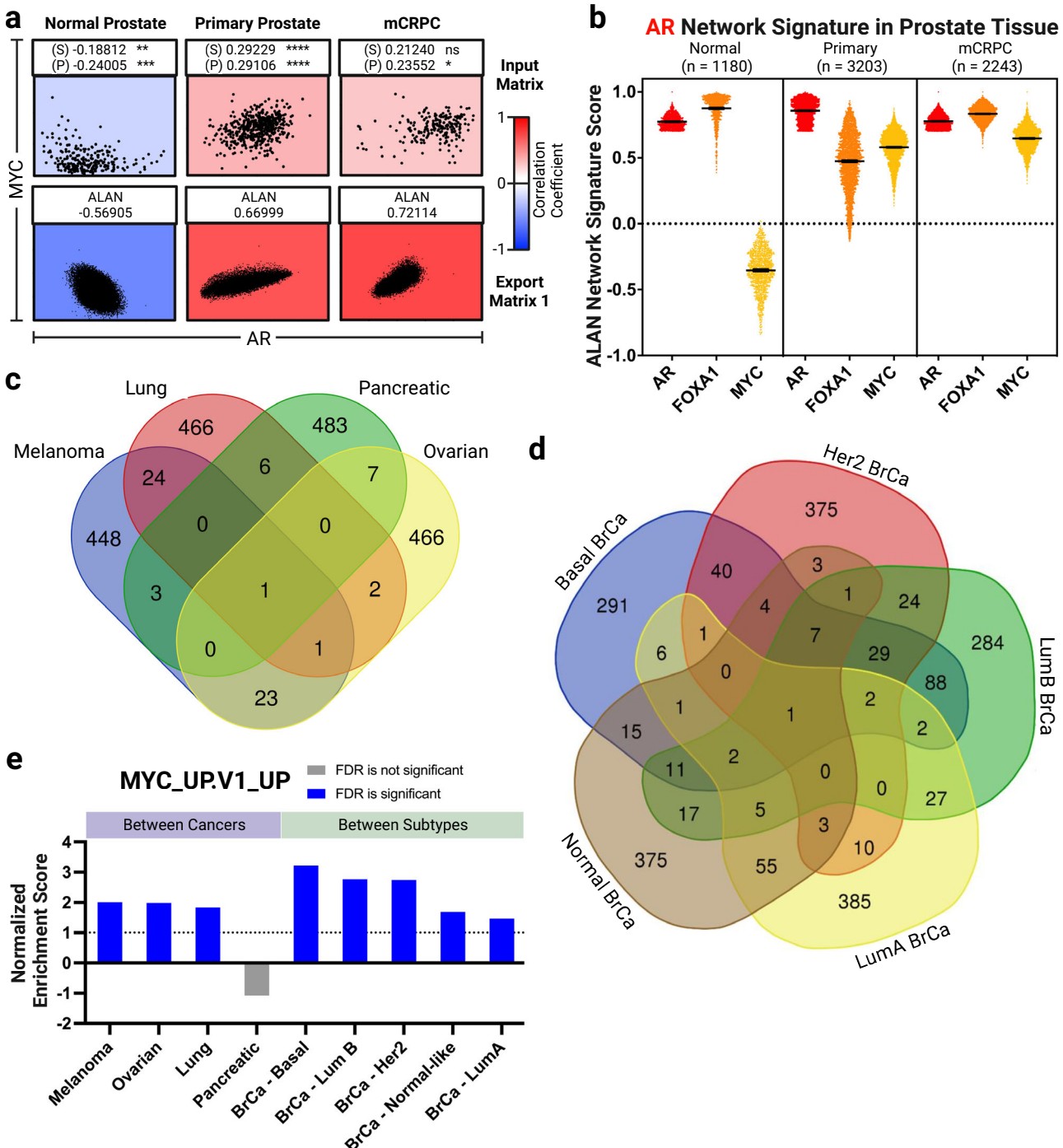

**Fig. 3 ALAN identifies context-specific gene behavior across prostate tissues, across different cancer types, and within cancer subtypes. a** The correlation between AR and MYC is depicted from blue (-1) to red (+1) in normal prostate tissue, primary prostate tissue, and mCRPC. Input matrix data was used for the top three visuals where each dot is a patient and its x- and y- coordinates represent the gene expression of *AR* or *MYC*, respectively. ALAN Output - Matrix 1 was used for the bottom three visuals where each dot represents an individual gene and its x- and y- coordinates represent the ALAN profile correlations with either *AR* or *MYC*, respectively. The numerical ALAN value between AR and MYC was obtained from ALAN Output – Matrix 2. **b** The ALAN profile correlation value is plotted for each gene within the ALAN *AR* network signature for the ALAN networks of *AR* (red), *FOXA1* (orange), and *MYC* (yellow). Individual genes in the indicated signature are compared in normal, primary, and mCRPC where the number of genes in each signature is indicated (n) and error bars represent mean with 95% confidence interval of the ALAN network signature score. **c** The overlap between *MYC* ALAN network signatures across four distinct cancer types in TCGA (Melanoma, Lung, Pancreatic and Ovarian) are visualized using a Venn diagram. **d** The overlap between *MYC* ALAN network signatures across five subtypes of breast cancer (Basal, Her2, LumA, LumB, and Normal) are visualized using a Venn diagram. **e** GSEA enrichment analysis with NES and FDR statistics of the MYC_UP.V1_UP gene set and the MYC ALAN Network from four distinct cancer types (Melanoma, Lung, Pancreatic and Ovarian) and five subtypes of breast cancer (Basal, Her2, LumA, LumB, Normal-like) from TCGA.

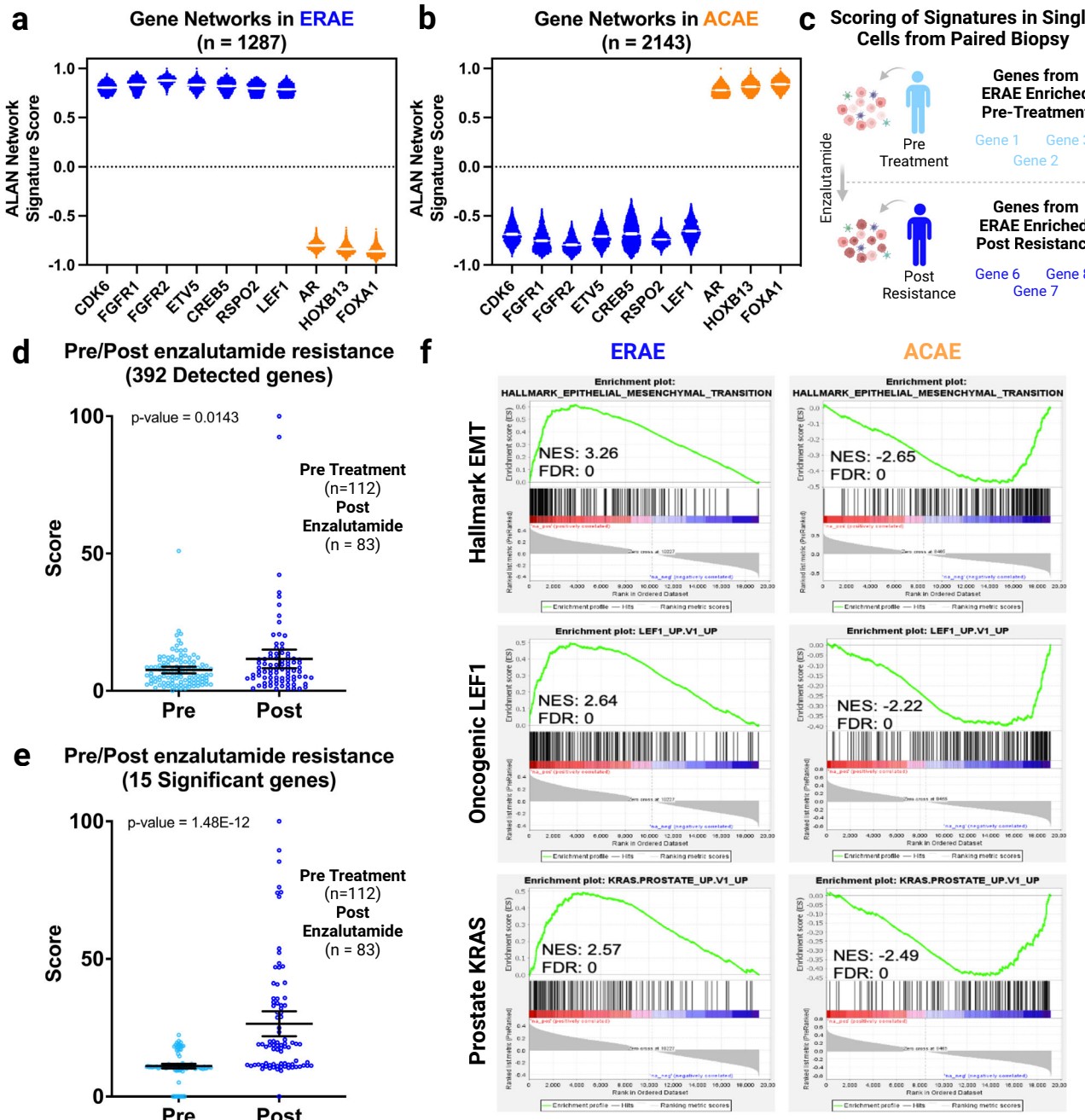

**Fig. 4 ALAN identifies resistant gene signature distinct from AR and co-factors that is enriched post enzalutamide treatment and associated with resistant oncogenic signatures. a** The ALAN profile correlation value is plotted for each gene within the ERAE for the ALAN networks in blue (*CDK6, FGFR1, ETV5, CREB5,* and *LEF1*) and orange (*AR, HOXB13,* and *FOXA1*). **b** The ALAN profile correlation value is plotted for each gene within the ACAE for the ALAN networks in blue (*CDK6, FGFR1, ETV5, CREB5,* and *LEF1*) and orange (*AR, HOXB13,* and *FOXA1*). For **a** and **b**, error bars represent mean with 95% confidence interval of the ALAN network signature score. **c** Workflow depiction of using a unique ALAN gene ecosystem signature for scoring cells from an individual patient with paired biopsies. **d** Scores are computed using the ERAE in all tumor cells that were pre-treatment (*n* = 112) or post-Enzalutamide resistant (*n* = 83) in one mCRPC patient. 392 of 1288 genes were detected in the scRNA-seq data. The statistical significance (p-value) was computed using a Student's t-test. **e** Scores are depicted for all tumor cells from **c** using the 15 genes with the greatest expression increases in the postresistant samples. For **d** and **e**, error bars represent mean with 95% confidence interval of the ERAE score. **f** Aggregate ERAE and ACAE are analyzed and depicted through GSEA Enrichment Plots. Hallmark EMT, Oncogenic *LEF1* and Prostate *KRAS* are shown with NES and FDR statistics.

of scRNAseq data, not all genes that were sequenced had detectable count values. Therefore, we first scored all genes that were detected in both paired biopsies from the ERAE (392 genes) and found a statistically significant increase across the tumor cells from the enzalutamide-resistant biopsy (Fig. 4d). We subsequently noted that 15 of the 392 genes demonstrated the most robust statistical significance (*p* value < 0.0001) in this patient

post treatment (Fig. 4e). This result demonstrates that ALAN is powered to generate unique gene signatures and potentially gene panels based on clinical response and resistance status. Additional studies are required to validate these findings in additional patients.

To further interrogate the biology associated with the ACAE and ERAEs in mCRPC patients, we performed GSEA and found

that gene signatures, including EMT, oncogenic *LEF1*, and several tissue-associated KRAS signatures (NES = 3.26, 2.64, 2.57; FDR = 0.0) were positively associated with the ERAE (Fig. 4f)[10]. Conversely, the ACAE exhibited negative enrichment of these same signatures. This result indicates that these two ALAN ecosystems have highly opposing functions based on conventional enrichment approaches, such as Gene Set Enrichment Analysis[20]. Since the ACAE constitutes of genes that regulate AR signaling, we also examined 83 genes that are representative of AR signaling, AR Nelson Up[47]. The 83 genes belonging to the AR Nelson pathway scored better than random in the ACAE (Supplementary Figure 2b). Reciprocally, genes in the ERAE, which have the opposing function, scored worse than random selection (Supplementary Figure 2b). These oppositional ecosystems further bolster our previous result (Fig. 2e) and indicates that ALAN identifies these groups of genes, using both gene networks and now gene ecosystems, as potentially separate mechanisms in mCRPC patients. Overall, the two ALAN gene ecosystems exhibited dichotomous relationships, as characterized by their underlying gene networks and signaling pathways.

**The resistance gene ecosystem nominates potential future therapeutic targets**. We also sought to further examine the ERAE to predict alternative targets that could be used against mCRPCs that develop resistance to ART. We prioritized surfacesome and secretome proteins for future antibody (neutralizing, radio-labeled, drug-conjugates) or immune-cell (Natural Killer, T-Cell engagers, CAR-T cells) therapies. Upon integrating ERAE and ACAE with proteins found in the Surfacesome[12,13] (Fig. 5a) or Secretome[13] (Fig. 5b), we found these ecosystems again included distinctly different proteins. To further illustrate the distinctiveness of the ERAE and ACAE, we depicted their relative degree of association with each other and all other genes in mCRPC via dimensional reduction on an x-y-plane through Uniform Manifold Approximation and Projection (UMAP, Fig. 5c). Notably, we identified *RSPO2* within the ERAE. Previous studies have demonstrated that mCRPCs harbor amplification or expression-driving fusion events of *RSPO2*, which encodes a secreted WNT signaling enhancer[25]. The *RSPO2* network overall had a positive ALAN network correlation with other enzalutamide-resistant gene networks, including *CDK6*, *LEF1* and *FGFR1* (Fig. 5d, all *p* value < 0.0001), while we observed an opposing ALAN network correlation within the *AR* ALAN network (Fig. 5e, *p* value < 0.0001). This result indicates that *RSPO2* is behaving more similar to genes within the ERAE than genes within the ACAE in mCRPC patients. We also confirmed the genomic observations of previous studies in which *RSPO2* is a recurrently amplified gene with limited deletions in prostate cancers (Fig. 5f). Further, in studies that sampled both primary and mCRPC patients, *RSPO2* amplifications were observed at increased rates in the mCRPC samples (Fig. 5g). Altogether, these findings indicate that ALAN ecosystems can be utilized to identify more genes with similar functional networks, and that this approach may yield targets that are on the surface or are secreted in mCRPCs.

## Discussion

The current scientific environment demands improved informatics-based approaches to research gene behavior. Existing tools utilize prior knowledge of signaling pathways and gene interactions to define their algorithms. In laboratory or informatics settings, we understand that gene behavior is regulated by other genes, as well as contexts, including cell type, micro-environment, treatment status, etc. However, no current tools have been developed to directly address all these differential regulatory features. In this study, we developed the Algorithm for

Linking Activity Networks (ALAN) as an algorithm to identify and compare the behavior of genes. ALAN first constructs gene ecosystems purely based on cohorts of patient data. ALAN outputs also include measurements of similarity between all possible gene networks, which allows for direct comparisons of gene behaviors in the same or distinct ecosystems. Since ALAN recognizes genes with similar behavior across multiple stages of a cancer, or even across distinct cell types, we demonstrate that ALAN can be utilized to identify relationships that exist outside of the static definition of gene pathways in cancer. We further utilized ALAN to construct gene ecosystems that promote therapy-resistant prostate cancer which identified sets of promising gene targets and signatures. The demonstration of these utilities indicate ALAN represents a mechanism to model cell signaling, protein-protein interactions, or gene behavior.

On a technical level, ALAN is compatible with many existing data types and platforms that include pipelines and pathway enrichment tools. While we have demonstrated that RNA and protein abundance level data can be utilized as ALAN input matrices, upstream pre-processing steps are relevant in minimizing technical artifacts. To this degree, users must still incorporate such tools towards generating the ALAN input matrix prior to implementation. Due to the rank-based conversion in its initial analyses, ALAN is agnostic to the shape of the data's distribution. If the two assumptions of the input matrix are met, then no further data transformation is required to run the ALAN algorithm (e.g., log transformation or normalization). As a result of this feature, ALAN circumvents the need for raw sequencing files such as BAM or FASTQ, which may not be readily available. Therefore, ALAN yields user-friendly intermediate and final output matrix files in tabular formats for users to subsequently review, interpret, and illustrate key findings with full customizability. Galaxy is one workflow manager in which users can identify gene-profiles, druggable targets, and relevant mutations in cancer[48]. ALAN can be incorporated into current Galaxy workflows after the generation of mapped expression profiles. The pairing of a traditional workflow that both analyzes raw sequencing data and consolidates relevant cancer information with ALAN's networking approach can ultimately deepen our understanding of how context-specific gene behavior can be applied to cancer treatment.

We have demonstrated that ALAN identifies genes with similar behaviors based on the expression patterns of all genes. We found that certain genes nominated by ALAN would not have been considered statistically significant if pairwise expression correlation approaches were utilized. Enrichment approaches, such as GSEA[20] can be applied to detect if a curated gene set is over-represented. We demonstrated that the ALAN gene networks are adaptable and nominate different genes when studying different cancer types and cancer subtypes. While GSEA is appropriate in determining if a known pathway is active, ALAN provides the users a tool that can nominate genes that behave similarly only in certain cancer types. This has utility when a user has interest in finding groups of genes that contribute to a signaling pathway in an unknown context, or in defining a pathway in which a gene set has not been devised. As an example, upon comparing signaling as functions of prostate cancer development, it was clear that some gene networks are consistently similar (i.e. *AR* and *FOXA1*) whereas others are highly dependent on the ecosystem (i.e. *MYC*). In situations where a disease biology is poorly understood, ALAN may be used to develop custom gene signatures for biomarkers that have critical clinical functions within that disease context.

Hierarchical clustering methods like ward.D2[49] and hierarchical tree-cutting tools, such as cutreeDynamic[50] use metrics of gene similarity to assign genes into distinct groups. The expression patterns of all genes contribute to the clustering and

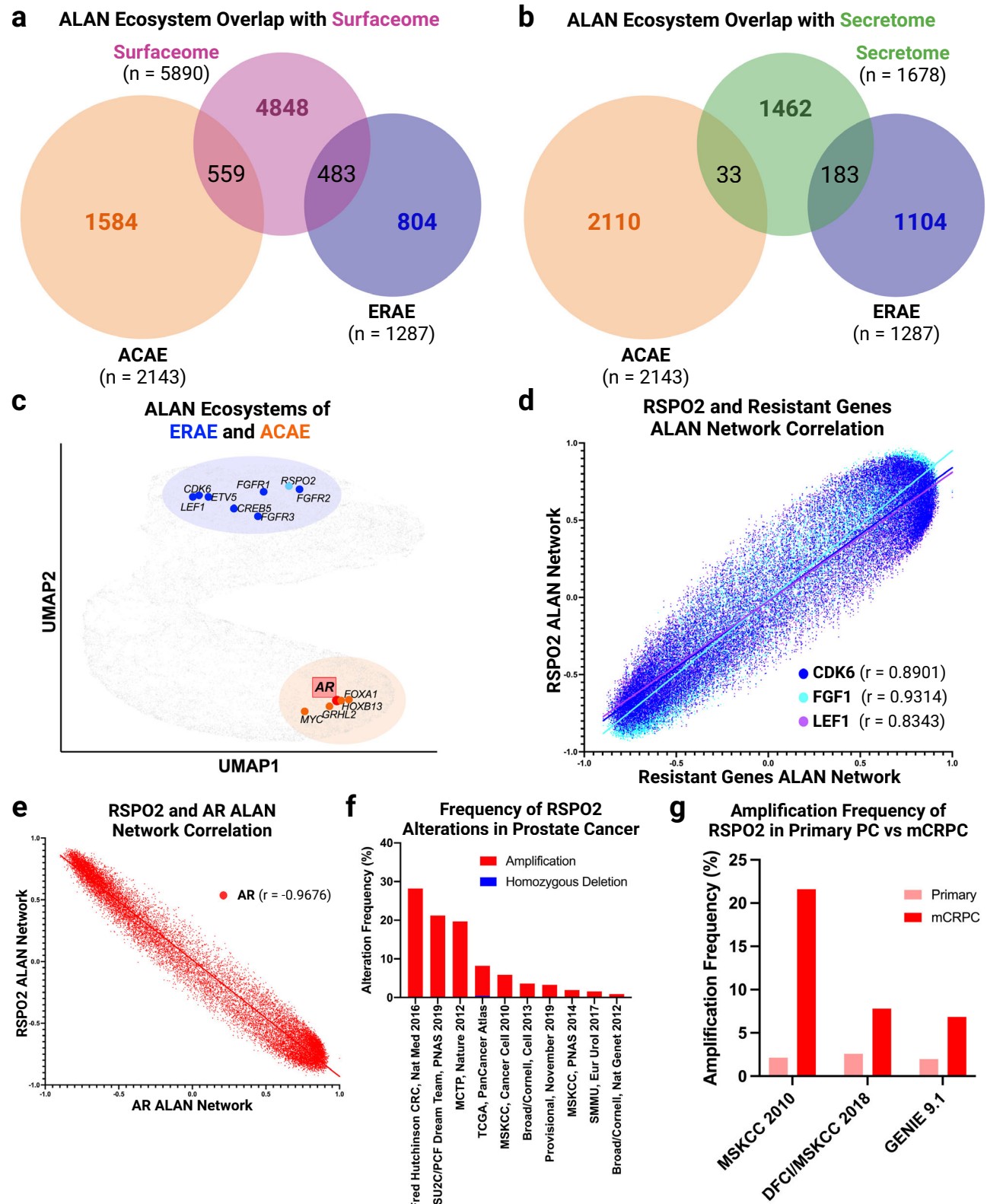

the outputs of clustering approaches are cluster labels for each element (gene). ALAN outputs do not inherently identify clusters of genes, but instead measure the similarity of all possible gene pairs based on numerical values. This provides an advantage to users that wish to visualize the relationship between a few specific genes. Users should consider the distinct utility of these parallel approaches as these functions may complement one another. As

one example, a user may utilize ALAN to depict why certain genes are in the same or separate gene clusters.

AI/ML tools are now available to study gene behavior with respect to outcomes. These tools classify genes based on specific features[21–23]. Here, ALAN outputs can be applied on the same datasets to determine if genes that fall in the same class do in fact exhibit similar gene behavior. In our prior study, we developed

**Fig. 5 ALAN nominates secreted protein RSPO2 as potential resistant gene via association with ERAE in advanced prostate tissue. a** The gene overlap of ERAE (blue) and ACAE (orange) with the Surfaceome are depicted using Venn Diagrams where the number of genes in each group (n) and the number of overlapping genes are depicted. **b** The gene overlap of ERAE (blue) and ACAE (orange) with the Secretome are depicted using Venn Diagrams where the number of genes in each group (n) and the number of overlapping genes are depicted. **c** UMAP plot depicting ALAN gene ecosystems in which specific gene networks of known biological activity are highlighted. Both the ERAE (blue) and ACAE (orange) are shown as well as the nominated resistant gene *RSPO2* (light blue). **d** The ALAN network correlation of the *RSPO2* network with ALAN networks for *CDK6* (blue), *FGFR* (light blue), and *LEF1* (purple) is depicted using ALAN cloud plots with linear regression statistics shown. **e** The ALAN network correlation of the *RSPO2* network with the *AR* (red) ALAN network is depicted using ALAN cloud plots with linear regression statistics shown. **f** Alteration frequency of *RSPO2* amplifications and homozygous deletions in various prostate cancer (PCA) datasets. **g** Comparison of *RSPO2* amplification frequency in primary PCA vs mCRPC patients in MSK2010, DFCI/MSKCC 2018, and GENIE 9.1 datasets.

P-NET to leverage cancer genomic data in order to predict critical classifiers that are potential biomarkers or even precision targets in metastatic prostate cancers[21]. The architecture of this neural networking consisted of a multi-layered hierarchical network structure of genes where their interaction was inferred based on 3007 curated biological pathways. Given that we were studying distinct states of prostate cancer (primary and metastatic), it is possible that these preconceived pathways do not reflect the true relationship of genes within primary and metastatic prostate cancers. In cases like P-NET, we can utilize ALAN to construct the hidden layers. While we propose that the use of ALAN may enhance AI/ML tools in studying gene behavior, it remains to be determined how these approaches can be integrated and how this will impact the results. Regardless, while numerous approaches can be used to compare gene behavior, strategies, including laboratory-based approaches are ultimately required to validate true biological functions.

Broadly, using ALAN to study gene networks may deepen understanding of how genes behave in distinct ecosystems, including different subtypes and prior exposure to therapeutics. In the molecular subtyping of breast cancer[19,51], expression-based approaches are applied to examine genes involving hormone sensitivity. Through observation of *ESR1*, *ESR2*, and *PGR* in Fig. 3 using ALAN, we have demonstrated that each gene has a distinct networking pattern across all subtypes of breast cancer. Given that positive expression of these genes is associated with more than one subtype, the ALAN result suggests that absolute expression patterns are not always indicative of the same phenomenon. Further investigation into how these differential gene networking patterns contribute to overall mechanisms within specific subtypes of breast cancer could enhance our understanding of subtype-specific gene behavior. Particularly, several RNA-based gene signatures have been developed as diagnostic or prognostic biomarkers to stratify prostate cancer patient outcomes. In tumor samples, these signatures inform the degree of *AR* activity (NEPC[52], AR Nelson[47]), enzalutamide resistance[46], or overall disease risk[17]. However, the gene list used in signatures were curated through recurrent laboratory findings or regression-based models. While they are without doubt co-expressed genes in specific settings, there may be limited understanding of the behavior of each gene within signatures. In addition, by virtue of applying the signature, we adapt the assumption that interrelationships of the genes within each signature persists in all contexts. The utilities of ALAN allow rapid modeling of gene programs that naturally occur in distinct contexts based on data that often already exists. This expands the development of context-specific signatures that may have improved predictive power for therapeutic response, survival, or other outcomes.

Investigating gene networks using ALAN networks may improve the detection of signal activation of genes in distinct ecosystems. This is critical towards the purposing or repurposing of cancer therapeutics and enhances investigations of therapeutic

sensitivity and patient prognosis. In biomarker studies, *BRCA1/2* dysregulations are used to stratify patients for the use of PARP inhibitors, but a subset of patients do not respond to this targeted treatment[53]. One can examine differential interactions within ALAN *BRCA1/2* networks in the ecosystems that consist of either responsive or non-responsive patients. In studying tumorigenesis and patient prognosis, users can leverage ALAN to study why losses or mutations in genes like *P53* contribute to tumorigenesis or poor survival in a cancer type or tissue-specific manner. Since ALAN allows for the investigation of the same gene in multiple ecosystems, there is also value in conducting analyses on other genes that are not oncogenes or tumor suppressors but have pleiotropic phenotypes, including chromatin and epigenomic modifiers, transcription factors, and immunoglobulins.

In summary, we created ALAN to be used as a computational tool that both creates and interprets gene networks in the context of gene ecosystems while maintaining the input data architecture. Given the data, ALAN can be leveraged to aid the development of precision biomarkers and to purpose or current therapeutics to additional patient populations.

## Methods

The ALAN algorithm accepts input m × n matrices with m molecular IDs (rows) and n sample IDs (columns), where each cell contains a measurement of any molecular abundance, such as gene expression or protein abundance. The input data must be in a format that allows for direct comparison between samples. The ALAN algorithm next serially conducts two statistical correlation operations in which all intermediate matrices are exported for quantitative comparisons and visual assessment. For the first correlation, a rank-based Spearman's correlation is performed between every pair of columns (molecular IDs) which generates the first correlation matrix, Matrix 1. This Spearman's Correlation alone measures the correlation of expression patterns of gene A to gene B across all samples. These relational profiles are further compared while building the second correlation matrix by performing a Pearson's correlation between every pair of columns (relational profiles) of Matrix 1 to generate Matrix 2. For ALAN, the Pearson's Correlation on top of the Spearman's correlation assesses the relationship between gene A and gene B based on their correlative relationship with all other genes. This additional correlation, as compared to a single correlation, is what we utilized when comparing the behavior of genes in the specified patient samples. The current ALAN algorithm is ran on the Minnesota Supercomputing Institute's (MSI) High-Performance Computing (HPC) Nodes at the University of Minnesota.

**ALAN.** The Algorithm for Linking Activity Networks (ALAN) is an algorithm that identifies and compares the behavior of genes. ALAN users must first identify the relevant omic datasets as well as the genes of interest. In this manuscript, we utilized ALAN outputs to compare the behavior of genes of interest within the context of the input data. We have focused on a few specific types of gene behavior, including co-regulators of a signaling pathway, protein-protein interactions, or any set of genes that function similarly.

**ALAN profile.** An ALAN gene profile results from the ALAN Output – Matrix 1. For each gene, this represents the expression pattern of one gene compared to another gene with respect to all patients. Comparisons between ALAN profiles are used to generate ALAN networks in ALAN Output - Matrix 2.

**ALAN network.** An ALAN gene network results from the ALAN Output - Matrix 2. For each gene, this represents the behavior of one gene with respect to all other genes detected in the dataset. Throughout the manuscript, we have compared the behavior of two or more genes by directly comparing their ALAN gene networks.

**ALAN ecosystem**. An ALAN ecosystem encompasses a set of genes with similar gene behavior as based on the ALAN Output – Matrix 2. In this manuscript, we have defined the ALAN ecosystem based on pairs of genes with an ALAN gene profile correlation above 0.7.

**ALAN inputs from cancer whole transcriptome sequencing and protein abundance data**. We included mapped expression profiles of the following studies as published on cBioPortal (log-transformed and z-score normalized): SU2C/PCF 2019 Metastatic Prostate Adenocarcinoma ($n = 208$)[10], TCGA Prostate Adenocarcinoma ($n = 493$), TCGA Breast Invasive Carcinoma (Basal $n = 171$, Luminal A (LumA) $n = 499$, Luminal B (LumB) $n = 197$, Her2 $n = 78$, and Normal $n = 36$), TCGA Ovarian Serous Cystadenocarcinoma ($n = 300$), TCGA Skin Cutaneous Melanoma ($n = 443$), TCGA Pancreatic Adenocarcinoma ($n = 177$), TCGA Lung Adenocarcinoma ($n = 510$). Transcription data from normal prostate tissue samples ($n = 245$) was obtained from the GTEx portal[11]. Protein abundance data from primary prostate tissue samples ($n = 76$) was also included[28]. The expression level data from these cohorts were analyzed through the ALAN algorithm.

**Individual patient analysis and scoring**. We identified one mCRPC patient from a prior study in which single cell RNA-sequencing was performed on paired biopsies from both before and after enzalutamide treatment[46]. Of the data, we examined transcripts per million (TPM) for genes in ALAN ecosystems and scored them in all tumor cells by summing the z-scores of the TPM values and scaling the aggregate sums from 0-100.

**Gene set enrichment analysis (GSEA)**. We conducted pre-ranked GSEA to depict enrichment of ALAN outputs using Hallmark gene sets and C6 oncogenic signatures to obtain net enrichment score (NES) based on FDR of 0.0[20]. The ranked list of genes with ALAN scores were obtained from either Matrix 1 or 2.

**GISTIC 2.0 calling of the AR focal amplicon**. We identified focal amplifications in 493 primary prostate adenocarcinomas from the TCGA Pan Cancer (http://firebrowse.org/) study to identify the genomic regions that harbor frequent amplifications in prostate cancer (PCA) including that of Xq12[29,30].

**HUGO (HGNC) mapping of chromosomal locations**. We utilized the HGNC Multi-Symbol Checker to identify the chromosomal locations of the 2244 genes in the AR network signature in the SU2C 2019 mCRPC study[10,54].

**Consolidation of surfaceome and secretome**. To create the list of surface proteins consolidated into the Surfaceome we integrated datasets from the Cell Surface Protein Atlas[12], which includes 1492 protein IDs and the Human Protein Atlas[13], which includes 5318 protein IDs, for a total of 5890 cell surface proteins. The list of 1678 secreted proteins used to define the Secretome was obtained from the Human Protein Atlas[13].

**Statistics and reproducibility**. Pairwise gene correlation coefficients (Spearman and Pearson's), p-values, and adjusted p-value statistics were calculated using R 4.2.2 (R Core Team; 2022), the stats (v4.2.2; R Core Team; 2022), the Tidyverse (v1.3.2; Wickham; 2019), and the Hmsic (v4.7-2; Harrell Jr F; 2022) packages.

**Inclusion and ethics**. All data included in this study was derived from public resources and these resources are provided in the manuscript. Ethical approval from the source datasets included can be found in the respective sources.

**Reporting summary**. Further information on research design is available in the Nature Portfolio Reporting Summary linked to this article.

## Data availability

Datasets derived from public resources. These resources are provided within the article and as follows. (1) The SU2C/PCF data are available at www.cbioportal.org (log transformed and z-score normalized) and in Dataset S1 of the original manuscript. https://doi.org/10.1073/pnas.1902651116 (2019)[10]. (2) The TCGA data (Prostate Adenocarcinoma, Breast Invasive Carcinoma, Ovarian Serous Cystadenocarcinoma, Skin Cutaneous Melanoma, Pancreatic Adenocarcinoma, and Lung Adenocarcinoma) are available at www.cbioportal.org (log transformed and z-score normalized) and at https://portal.gdc.cancer.gov/. (3) The GTEx data are available through the GTEx portal (www.gtexportal.org). https://doi.org/10.3390/jpm5010022 (2015)[11]. (4) For the single-cell sequencing (scRNA-seq) dataset, scRNA-seq expression data are available in Supplementary Data of the original manuscript with cellular annotations located in Supplementary Tables 1-6. Raw sequence data generated in this study are being deposited in dbGaP (accession phs001988.v1.p1). https://doi.org/10.1038/s41591-021-01244-6 (2021)[46]. (5) The protein abundance data from primary prostate tissue samples used in Supplementary Figure 1a are available in Supplementary Information Table S2 of the original manuscript. https://doi.org/10.1016/j.ccell.2019.02.005 (2019)[28]. The data used to generate the figures in this manuscript are available in Supplementary Data 1.

## Code availability

The basic ALAN operation source code has been deposited in a Zenodo Digital Repository https://doi.org/10.5281/zenodo.7770301 (2023)[55].

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

## Acknowledgements

We thank our funding sources Ray of Light Foundation (University of Minnesota) and American Cancer Society in supporting this work. Ray of Light Foundation award from Division of Hematology, Oncology, and Transplantation, University of Minnesota was allocated to the research efforts of J.H. and H.E.B. American Cancer Society (IRG-21-049-61-IRG) was allocated to the research efforts of J.H. and S.T. The figures and workflows shown here were created with https://biorender.com/. The results shown here are in whole or part based upon data generated by the TCGA Research Network: https://www.cancer.gov/tcga.

## Author contributions

H.E.B. and J.H. conceived the project. H.E.B. contributed to the design and implementation of the ALAN algorithm, design and analysis corresponding to all figures, and the writing, edits, revisions, and submission of the full manuscript. A.S. contributed to the design and implementation of the ALAN algorithm and the writing of the sections on AI/ML. A.D. contributed to the design and implementation of the ALAN algorithm, the writing of the method section, and conceptualization of the revision comments. A.A. and E.B. contributed to the statistical writing and the implementation of the hypergeometric distribution included in the revision comments and revised manuscript. S.T. contributed to the investigation of RSPO2 in mCRPC and the editing of the final manuscript document. J.R.L. contributed to the editing of the final manuscript document and the conceptualization of the revision comments. X.S., C.P.K., S.M., M.L., J.M.D. S.M.D. C.J.R. participated in the study design. S.P.P. contributed to the visualization of the revised Fig. 3A that was included in the revision comments and the final MS. E.S.A. conceptualized the clinical implications of the ALAN algorithm. J.W. contributed to the design of the ALAN algorithm. J.H., as the corresponding author on this manuscript, contributed to the design of the ALAN algorithm, the conceptualization of all analyses, and the writing, editing, and revisions of the final manuscript. All authors contributed to the editing of the final manuscript document.

## Competing interests

E.S.A. is a paid consultant/advisor to Janssen, Astellas, Sanofi, Dendreon, Pfizer, Amgen, Eli Lilly, Bayer, AstraZeneca, Bristol Myers Squibb, ESSA, Clovis, Merck, Curium, Blue Earth Diagnostics, Foundation Medicine, Exact Sciences and Invitae; has received research funding to his institution from Janssen, Johnson & Johnson, Sanofi, Dendreon, Genentech, Novartis, Tokai, Bristol Myers Squibb, Constellation, Bayer, AstraZeneca, Clovis and Merck; and is the coinventor of a patented AR-V7 biomarker technology that has been licensed to Qiagen. J.H is a paid consultant/advisor to Caris Life Sciences and Astrin Biosciences J.M.D. is Chief Scientific Officer and owns stock options at Astrin Biosciences. The other authors declare no competing interests.
