## [Peer Review File · Communications Biology]

Reviewers' comments:

Reviewer #1 (Remarks to the Author):

This study utilized a novel analytical method called ALAN to compare behavior of genes in ecosystems. Through this method, they identified novel gene signatures, gene targets, and mechanisms of progression or resistance seen in cancer patients. ALAN could be useful in dissecting context specific gene behaviors based on gene expression data. Overall, the figures are well crafted, and the manuscript is well written. My major concern is regarding the rationale and design of the ALAN analysis, and its advantage over standard gene clustering/correlation methods:

Major comments:

- It is unclear why Pearson Correlation is performed on top of the Spearman's Correlation matrix between genes. In this case, Pearson correlation means how much does gene A correlate with gene B based on their expression correlations with all other genes. Do genes within the same "ecosystems" share more protein interactions and pathways than the gene clusters defined directly based on the spearman correlations matrix?
- The conventional definition of ecosystems means that these genes are interacting with each other. It is unclear why matrix 2 (based on Pearson correlation) is defined as gene ecosystems provided that there is no evidence that they interact with each other at protein level.
- There is a lack of assessments of functional relations between genes within the same ecosystem. For example, do genes within the same ecosystem share more protein interactions and pathways than randomly selected genes of the same numbers?
- It is unclear what is the advantage of the ALAN analysis over standard gene clustering/correlation analysis. A comparative analysis of ALAN analysis with standard gene clustering methods (i.e., using "ward D2" clustering on the gene expression data, and then use "cutreeDynamic" to identify gene groups), and gene correlation methods (i.e., Pearson Correlation of gene expression) should be carried out. The benchmark may be carried out based on the functional associations between the genes within the same ecosystem/gene cluster, as indicated by their average number of shared molecular interactions and pathways.

Reviewer #2 (Remarks to the Author):

Bergom et al., present here an Algorithm for Linking Activity Networks (ALAN) computational pipeline for a better understanding of context-specific cancer therapy. Applying three cancer types, mainly focusing on prostate cancer were used to present to show the application of the ALAN computational pipeline. The authors were able to recapitulate several known associations or biomarkers in prostate or breast cancer. While their questions are interesting, there are several concerns that should be addressed prior to publication:

Major comments:

- Introduction

The authors pointed out the current limitations of ML/AI-based algorithms, especially focusing on PNET.

On page 2: "We previously applied PNET, a neural-networking algorithm, to identify somatic driver events using DNA-sequencing data from primary prostate cancer and mCRPC. We identified one genomic feature, MDM4 amplifications, that represented a promising precision therapy target in mCRPC. The architecture of PNET was based on 3,007 pre-curated biological pathways that were not cancer-type specific. Overall, while these AI/ML approaches predict risk of occurrence from tumor samples, they include user/machine bias. This is because prior to analyses, users must pre-define the selectable features in order to build the model's architecture. The selectable features are often based

on known gene signatures and static definitions of biological pathways. For these reasons, the interoperability and outcomes are highly contingent on pathway structures, which will be inaccurate when contexts change. Overall, an informatic driven solution that considers the context specificity could enhance the learning capacity of AI/ML tools."

This part needs to improve for a more logical introduction. The authors need to clarify these points:

Q: Are there any other ML/AI-based algorithms beyond PNET? Can we generalize the limitations only using the PNET example?

Q: It seems like even PTEN included many different biological features such as copy-number changes, gene expression level, and mutations (from Fig1 in the original paper). What is the difference or strong point of ALAN method compared to PNET exactly?

- Algorithm for Linking Activity Networks (ALAN) computational pipeline

There are many uncertain parts that need to address.

Q: on page2, "where each cell contains a measurement of any molecular abundance such as gene expression, counts, or protein abundance" It is unclear whether ALAN is allowed to use any kind of expression (or abundance data) either gene expression or protein abundance. If yes, could the authors also validate or provide the possibility to use it in the protein abundance data?

Q: on page2, "Entirely dependent on the input data, ALAN acts as an informatic-based assay that measures contextual gene behavior."

It seems like this method is mainly used gene expression data. It is not so clear how a single feature (gene expression) could perfectly measure context-specific behaviors. If this is true, comparisons between different cancer types will be necessary to prove it.

Q: on page3, "The ALAN pipeline next serially conducts two statistical correlation operations in which all intermediate matrices are exported for quantitative comparisons and visual assessment. A rank-based Spearman's correlation is then performed between every pair of columns (molecular IDs) which generates the first correlation matrix, Matrix 1. These relational profiles are further compared while building the second correlation matrix by performing a Pearson's correlation between every pair of columns (relational profiles) of Matrix 1 to generate Matrix 2. This subsequent correlation matrix provides a comparative functional network profile between molecular variables defined as the gene ecosystem of the input data."

It is not so clear the purpose of the first and second steps and what's the difference between the two steps. Also, Figure 1 is not clearly present the difference between the two steps.

Q: on page3, "ALAN pipeline without any pre-processing or normalization." The authors need to clarify why they do not need any pre-processing or normalization steps? Or how do they handle removing technical artifacts from different sources? If input samples include technical artifacts, are there any filtering steps to resolve them? It is also not clear which TCGA samples they used here since TCGA provided gene expression data either raw data and processed or normalized data.

- Cancer type selection

This work is mainly applied to prostate cancer and also two cancer types (breast cancer and ovarian cancer). Since the method is basically designed to use gene expression data, it is not clear why the authors applied their method to three specific cancer types. Especially, TCGA is composed of more than 30 cancer types, this method would be expected to use in a general manner. Could you clarify the definition of selected cancer types or apply this method to other cancer types?

Also, it is not clear why the method description and analysis are differently applied to prostate cancer and breast or ovarian cancer. In the method section: GISTIC 2.0 calling of the AR focal amplicon (on page 4): only for prostate cancer

Transcription data from normal prostate tissue samples (n = 245) was obtained from the GTExportal (on page 3): only for prostate cancer

Additionally, in the discussion section, "Since ALAN recognizes all intricate gene-to-gene interactions across multiple stages of a cancer, or even across distinct cell types, we demonstrate that ALAN avoids the conventional and static definition of gene pathways in cancer." The authors generalized the application of the method. However, it is not so sure about multiple stages of a cancer or even across distinct cell types.

- Figure

Figure1: in the top panel, there are two boxes indicating sample collection steps experimentally. However, this method is designed to use publicly available transcriptomics data sets and sample collection steps were not done in this manuscript. It is better to delete it. in the bottom panel, data processing is in the middle but it makes a misunderstanding of the order of process. 'Data processing' step would be better to be the first. The 'connectivity' panel is difficult to understand. What is the color/thickness/dashed line indicated? Also, the gene network/ecosystem should be defined in the figure more clearly.

Minor comments:

1. In several sentences, references were not correctly addressed. For example, - We 282 examined scRNA seq data obtained from paired biopsy samples from one patient pre-enzalutamide treatment and 283 post-therapy resistance (on page6: Authors need to clarify which data set are you referring to here).

- Many studies have accrued -omic data from tumor-derived specimens, including DNA (genomics), RNA (transcriptomics), protein/phosphoprotein (proteomics/phosphoproteomics), epigenetics (epigenomics), or even metabolites (metabolomics) [on page2: Authors need to point out details examples]

- AI and ML also have high potential to enhance information obtained from molecular diagnostic 69 assays by deconvoluting relationships between patient outcomes, genes, and signaling pathways (on page2: need to address current articles based on AI/ML).

2. What is the correct name of PNET or P-NET? Because the name in the original paper is P-NET not PNET.

3. In this manuscript, some terminologies were commonly used without clear definitions such as ecosystem, gene network, gene interaction, or gene-gene interactions. It makes it difficult to understand the manuscript since the meaning is not perfectly matched with the common definition in the field. For example, gene interaction usually indicates genetic interaction that could be a co-occurring or mutually exclusive relationship. Therefore, the authors need to clarify them correctly. There are several examples that need to improve.

- On page 2, While current informatics tools "classify" signaling pathways and biomarkers, understanding of gene behavior is an entirely distinct scientific objective that requires a tool that evaluates biological activities of the same gene in multiple contexts. Gene networks represent a mechanism to measure how one gene behaves. -> There is no definition of the gene network. Also, the network usually indicates the overall behavior, not a single (one) gene.

- On page2, We can represent gene networks as the summation of the varying degrees of all gene-to-gene interactions using RNA-sequencing data across many patient samples.

- On page7, allows direct comparisons of informatics-based phenotypes -> what are informatics-based phenotypes?

4. Several sentences are difficult to understand due to the inconsistent description. For example. On

page 4, "We first tested the functionality of ALAN to interpret projected interactions between AR and known co-factors in mCRPC patients, of which many are intrachromosomal. To map co-amplified genes of AR, we utilized outputs from GISTIC 2 to identify the recurrent focally amplified region of AR in 492 primary prostate cancer tumors (Figure 2A). AR resides in a focal amplicon on Xq12 with 9 additional genes where only 8 of the 9 genes were detected in this cohort. We first examined if ALAN could predict known AR interactions and if genomic interactions, such as focal amplifications, were indicative of similar gene profiles and gene networks in mCRPC patients."Here, it is not so clear which data sets they used here between mCRPC patients and primary prostate cancer tumors.

Detailed Response to Reviewers.

We sincerely thank the Reviewers for the constructive critiques. We have extensively revised the manuscript based on these comments. This has greatly improved our study.

Central to the comments from both reviewers, we have clearly defined ALAN as an algorithm distinct from AI/ML, clustering, or correlation approaches. Given that we have used specific terminology in the manuscript, we recognize the relevance of clarifying the definitions of gene ecosystems and gene behaviors. Our use of the following terminology is specific to ALAN outputs.

To facilitate the review of our revisions, below are specific definitions of key terms used in the manuscript.

ALAN: The Algorithm for Linking Activity Networks (ALAN) is an algorithm that identifies and compares the behavior of genes. ALAN users must first identify the relevant omic datasets as well as the genes of interest. In this manuscript, we utilized ALAN outputs to compare the behavior of genes of interest within the context of the input data. In the revised manuscript, we have focused on a few specific types of gene behavior including co-regulators of a signaling pathway, protein-protein interactions, or any set of genes that function similarly (See **Reviewer 1 Comment 1**).

ALAN Network: An ALAN gene network results from the ALAN output. For each gene, this represents the behavior of one gene with respect to all other genes detected in the dataset. Throughout the manuscript, we have compared the behavior of two or more genes by directly comparing their ALAN gene networks. Because the ALAN network also accounts for the relationship between the genes of interest with all other genes, these comparisons are more robust than comparing the expression levels of two genes alone (See **Reviewer 1 Comment 1** and **Reviewer 2 Comment 2**).

ALAN Ecosystem: An ALAN ecosystem encompasses a set of genes with similar gene behavior as based on the ALAN outputs. In this manuscript, we have defined the ALAN ecosystem based on pairs of genes with an ALAN gene profile correlation above 0.7. We have elected this threshold based on both mathematical and biological justifications: correlation coefficients above 0.7 are extremely strong correlations and this threshold identified genes that are known to behave similarly based on literature.

Reviewer #1 (Remarks to the Author):

This study utilized a novel analytical method called ALAN to compare behavior of genes in ecosystems. Through this method, they identified novel gene signatures, gene targets, and mechanisms of progression or resistance seen in cancer patients. ALAN could be useful in dissecting context specific gene behaviors based on gene expression data. Overall, the figures are well crafted, and the manuscript is well written. My major concern is regarding the rationale and design of the ALAN analysis, and its advantage over standard gene clustering/correlation methods:

1. It is unclear why Pearson Correlation is performed on top of the Spearman's Correlation matrix between genes. In this case, Pearson correlation means how much does gene A correlate with gene B based on their expression correlations with all other genes. Do genes within the same "ecosystems" share more protein interactions and pathways than the gene clusters defined directly based on the spearman correlations matrix?

We thank **Reviewer 1** for these comments that yielded relevant analyses critical to the revised manuscript.

Indeed, the Spearman's Correlation alone first measures the correlation of expression patterns of gene A to gene B across all samples. For ALAN, the Pearson's Correlation on top of the Spearman's correlation assesses the relationship between gene A and gene B based on their relative relationship with all other genes (their ALAN networks). We highlight the functionality of this additional correlation, as compared to a single correlation in **Supplementary Tables 1-3** below.

To address the second part of the question, we have highlighted three forms of functional relationships shared by genes we found to be within an ALAN ecosystem, two of which are direct interactions at the protein level.

1. Protein Interactions. Through ALAN outputs based on data from metastatic castration resistant prostate cancers (mCRPC)¹, we found that AR, FOXA1, and HOXB13 are in the same ALAN ecosystem. We have cited studies that demonstrate they bind to one another and interact at the same transcription regulatory sites to promote Androgen Receptor (AR) signaling in prostate cancer^{2,3}. As shown, we performed a single pairwise gene correlation analysis of gene expression levels to compute the correlation coefficients between AR and FOXA1 or HOXB13 across all mCRPC patients (**Supplementary Table 1**). In comparing these to ALAN outputs, we note that the ALAN outputs were higher as compared to Spearman's or Pearson's correlation coefficients.

Supplementary Table 1. Protein Interactions

Gene1_Gene2	Spearman's Correlation	Pearson's Correlation	ALAN Export Matrix 2
AR_FOXA1	0.49750 ****	0.64749 ****	0.91655
AR_HOXB13	0.43934 ****	0.64218 ****	0.89588

adj p-value > 0.05 (ns), adj p-value < 0.05 (*), adj p-value < 0.01 (**), adj p-value < 0.001 (***), adj p-value < 0.0001 (****)

2. Co-regulators of a signaling pathway. We have cited prior work that indicates AR and MYC are co-regulators of signaling in prostate cancer⁴⁻⁶. We thus expected that AR and MYC would have similar behavior in only primary prostate cancer (TCGA) and mCRPC, but not in normal prostate tissue⁷. The ALAN outputs supported this biological relationship to a greater degree as compared to pairwise Spearman's or Pearson's correlation coefficients (**Supplementary Table 2**). Notably, as compared to Spearman's or Pearson's correlations, the ALAN outputs indicated that AR and MYC behave similarly in primary prostate cancer and mCRPC. Further, AR and MYC share an ALAN ecosystem in mCRPC. To note, through Spearman or Pearson correlations, AR and MYC had limited statistical significance in mCRPCs when only comparing the expression levels of the gene pairs.

Supplementary Table 2. Co-Regulators of the Same Pathway

Gene1_Gene2	Cohort	Spearman's Correlation	Pearson's Correlation	ALAN Export Matrix 2
AR_MYC	Normal Prostate	-0.18812 **	-0.24005 ***	-0.56905
AR_MYC	Primary Prostate	0.29229 ****	0.29106 ****	0.66999
AR_MYC	mCRPC	0.21240 ns	0.23552 *	0.72114

adj p-value > 0.05 (ns), adj p-value < 0.05 (*), adj p-value < 0.01 (**), adj p-value < 0.001 (***), adj p-value < 0.0001 (****)

3. Genes with similar functions. We have cited literature that independently established FGFR1, CREB5, and CDK6 as genes that promote resistance to AR targeted therapies in mCRPC⁸⁻¹⁰. Particularly, the family of FGFs proteins are active in mCRPCs that lack AR activity⁹. We thus expected a reciprocal relationship with AR and FGFs in mCRPC. In addition, of the proteins with known resistance functions (FGFR1, CREB5, and CDK6), we expected ALAN would yield outputs supporting that they have similar gene behavior. ALAN indeed identified positive correlations between FGFR1, CREB5, and CDK6 in mCRPC (**Supplementary Table 3**) and negative correlations with AR. Pairwise gene expression correlations were not as robust as the ALAN outputs. Notably,

while FGFR1, CDK6, and CREB5 are genes that promote resistance to therapies, the ALAN results also indicate they potentially interact with many of the same genes. This is a novel perspective that has not been reported in literature. To note, based on Spearman's or Pearson's correlations, we see limited statistical significance when only comparing the expression levels of the gene pairs. This highlights the difference of ALAN, as ALAN accounts for how these genes interact with all other genes that are detected.

Supplementary Table 3. Genes with Similar Function.

Gene1_Gene2	Spearman's Correlation	Pearson's Correlation	ALAN Export Matrix 2
FGFR1_CDK6	0.29616 ***	0.27019 **	0.82279
FGFR1_CREB5	0.27130 **	0.18176 ns	0.79487
CDK6_CREB5	0.32726 ****	0.21271 ns	0.82464
AR_CDK6	-0.18119 ns	-0.17542 ns	-0.75901
AR_CREB5	-0.28005 **	-0.24553 *	-0.76170
AR_FGFR1	-0.27272 **	-0.26861 **	-0.78961

adj p-value > 0.05 (ns), adj p-value < 0.05 (*), adj p-value < 0.01 (**), adj p-value < 0.001 (***), adj p-value < 0.0001 (****)

To reflect these revisions in the revised manuscript, we have conducted the following:

We clarified our definition of gene ecosystems in the following sections:

Materials and Methods section (page 3, lines 143-145), “*ALAN Ecosystem*: An ALAN ecosystem encompasses a set of genes with similar gene behavior as based on the ALAN Output – Matrix 2. In this manuscript, we have defined the ALAN ecosystem based on pairs of genes with an ALAN gene profile correlation above 0.7.”

Results section (page 4, lines 176-177), “Genes with highly similar networks as indicated by a gene profile correlation score of above 0.7 share an ALAN gene ecosystem.”

We have made additional revisions to the text in the following sections:

Introduction section (page 2, line 89-91), “The Algorithm for Linking Activity Networks (ALAN) is an algorithm that identifies and compares the behavior of genes. ALAN users must first identify the relevant clinical context, the associated omic datasets, as well as the genes of interest. ALAN is then used to compare the behavior of two genes of interest based on the context of the input data.”

Materials and Methods section (page 3, lines 102-103), “This Spearman's Correlation alone first measures the correlation of expression patterns of gene A to gene B across all samples.”

Materials and Methods section (page 3, line 105-108), “For ALAN, the Pearson's Correlation on top of the Spearman's correlation assesses the relationship between gene A and gene B based on their relative relationship with all other genes. This additional correlation, as compared to a single correlation, is what we utilized when comparing the behavior of genes in the specified patient samples.”

Results section (page 5, lines 240-247), “Previous literature has indicated that AR and MYC become co-regulators in prostate cancer, but only as a function of disease progression³⁵⁻³⁷. We thus expected that AR and MYC would have convergent signaling in prostate cancer and mCRPC, but to lesser degrees in normal prostate tissue. Upon analyzing AR and MYC in each clinical setting, the ALAN outputs robustly supported this biological relationship and nominated AR and MYC in the same ALAN ecosystem in mCRPC (**Figure 3A, Supplemental**

Table 2). Notably, pairwise Spearman’s or Pearson’s correlations of AR and MYC expression levels yielded limited significance when ALAN outputs were robust. These results demonstrate that ALAN outputs, which accounts for relationships of a gene with all others, reflect a cancer stage specific functional relationship between AR and MYC.”

Updated **Figure 3A.**

Figures and Legends section (page 12, lines 494-499), “**Figure 3.A.** The correlation between AR and MYC is depicted from red (-1) to blue (+1) in normal prostate tissue, primary prostate tissue, and mCRPC. Input matrix data was used for the top three visuals where each dot is a patient and its x- and y- coordinates represent the gene expression of AR or MYC respectfully. ALAN Export Matrix 1 was used for the bottom three visuals where each dot represents an individual gene and its x- and y- coordinates represent the ALAN profile correlations with either AR or MYC respectfully. The numerical ALAN value between AR and MYC was obtained from ALAN Output – Matrix 2.”

We revised **Figure 1. Applying the Algorithm for Linking Activity Networks (ALAN).**

Figures and Legends section (page 10, lines 466-468), “**Figure 1. Applying the Algorithm for Linking Activity Networks (ALAN).** Workflow depicting the ALAN algorithm which includes input matrix, matrix generation, user generated depictions of ALAN data matrices and key terms.”

Supplemental Tables 1-3 were added to the revised manuscript (page 17, lines 554-565) and the following textual revisions were added to the **Results** section:

- (page 5, lines 208-209), “ALAN outputs were robust compared to pairwise Spearman or Pearson’s correlations (**Supplemental Table 1**).”
- (page 5, lines 242 - 244), “Upon analyzing AR and MYC in each clinical setting, the ALAN outputs robustly supported this biological relationship and nominated AR and MYC in the same ALAN ecosystem in mCRPC (**Figure 3A, Supplemental Table 2**).”
- (page 6, lines 287 – 289), “Lastly, ALAN considered these genes to behave similarly, while pairwise Spearman or Pearson’s correlations often yielded insignificant comparisons (**Supplemental Table 3**).”

2. The conventional definition of ecosystems means that these genes are interacting with each other. It is unclear why matrix 2 (based on Pearson correlation) is defined as gene ecosystems provided that there is no evidence that they interact with each other at protein level.

We appreciate the comment by **Reviewer 1**.

As we are unaware of the conventional definition of gene ecosystems in our community, we have instead revised and clarified our own definition of the ALAN gene ecosystem (1st paragraph in **Detailed Response to Reviewers**). We have also revised the manuscript to reflect that genes in an ALAN ecosystem exhibit three forms of functional relationships, two of which are direct interactions at the protein level (**Reviewer 1 Comment 1**).

3. These is a lack of assessments of functional relations between genes within the same ecosystem. For example, do genes within the same ecosystem share more protein interactions and pathways than randomly selected genes of the same numbers?

We appreciate **Reviewer 1** for this comment that has improved the revised manuscript.

We have highlighted three forms of functional relationships between genes within the same ALAN ecosystem in **Reviewer 1 Comment 1**.

To examine the relationship between genes in the same ecosystems as compared to “random” genes, we elect to highlight two ALAN ecosystems we have discussed in the manuscript, the ERAE (Enzalutamide Resistant ALAN Ecosystem) and the ACAE (AR and Co-factors ALAN Ecosystem). We examine these ecosystems as follows:

1. These ecosystems have highly opposing functions based on conventional enrichment approaches, such as Gene Set Enrichment Analysis¹¹ (GSEA, **Figure 4E**). Specifically, this was previously depicted for the three pathways that were highly enriched in the ERAE (Hallmark EMT, Oncogenic *LEF1*, and Prostate *KRAS*) but were de-enriched in the ACAE.
2. Based on what we originally depicted in **Figure 4A**, there are “zero” genes that are overlapping between the ERAE (n = 1287) and the ACAE (n = 2143). As part of this revision, we have conducted a hypergeometric test, using a normal approximation to the hypergeometric distribution, this result of zero overlapping genes given these two groups is highly significant (p-value = 1.32e-69) indicating that this result was not due to chance. The expected number of overlapping genes based on chance is 144 given these two groups.
3. Since ACAE constitutes of genes that regulate AR signaling, we also examined 83 genes that are representative of AR signaling, AR Nelson Up¹². Based on the genes that constitute the ACAE and ERAE, we would expect that selection of these 83 genes to be greater than random based on the ACAE and worse than random based on the ERAE. The 83 genes belonging to the AR Nelson pathway scored better than random in the ACAE (**Supplemental Figure 2A**). Reciprocally, genes in the ERAE, which have the opposing function, scored worse than random selection (**Supplemental Figure 2B**).

To reflect these changes in the revised manuscript, we have included the following updates:

Results section (page 6, line 284-287), “To determine the significance of this result, we have conducted a hypergeometric test, using a normal approximation to the hypergeometric distribution. This result of zero overlapping genes given these two groups is highly significant (p-value = 1.32e-69) indicating that this result was not due to chance. The expected number of overlapping genes based on chance is 144 given these two groups.”

Results section (page 6, lines 305-307), “This result indicates that these two ALAN ecosystems have highly opposing functions based on conventional enrichment approaches, such as Gene Set Enrichment Analysis²⁰.”

Updates to include **Supplemental Figure 2. A and B**.

A. Ranking Accuracy of ALAN ACAE in AR Nelson

B. Ranking Accuracy of ALAN ERAE in AR Nelson

Supplemental Figures and Legends Section (pages 16 and 17, lines 546-552), “**Supplemental Figure 2.** Ranking Accuracy of ALAN Ecosystems against AR Nelson Gene Signature where performance against random ranks was measured by calculating the area under the curve (better > 0.5; worse < 0.5). **A.** The ACAE gene network represents the average gene networks of AR, FOXA1, and HOXB13 (ALAN Export Matrix 2). The resulting gene profile correlation coefficients were ranked and then the AR Nelson gene signature was scored. **B.** The ERAE gene network represents the average gene networks of CDK6, FGFR1, FGFR2, ETV5, CREB5, and LEF1 (ALAN Export Matrix 2). The resulting gene profile correlation coefficients were ranked, and then the AR Nelson gene signature was scored.”

Results section (page 6, lines 307-310), “Since the ACAE constitutes of genes that regulate AR signaling, we also examined 83 genes that are representative of AR signaling, AR Nelson Up⁴⁸. The 83 genes belonging to the AR Nelson pathway scored better than random in the ACAE (**Supplemental Figure 2A**). Reciprocally, genes in the ERAE, which have the opposing function, scored worse than random selection (**Supplemental Figure 2B**).”

4. It is unclear what is the advantage of the ALAN analysis over standard gene clustering/correlation analysis. A comparative analysis of ALAN analysis with standard gene clustering methods (i.e., using "ward D2" clustering on the gene expression data, and then use "cutreeDynamic" to identify gene groups), and gene correlation methods (i.e., Pearson Correlation of gene expression) should be carried out. The benchmark may be carried out based on the functional associations between the genes within the same ecosystem/gene cluster, as indicated by their average number of shared molecular interactions and pathways.

We appreciate **Reviewer 1** for these comments and the suggestion of a benchmark analysis against clustering approaches.

While both clustering approaches (ward.D2¹³, cutreeDynamic¹⁴) and ALAN utilize expression matrices to assess the similarity of genes, ALAN should be considered an orthogonal approach. ALAN does not inherently group genes into clusters but instead provides numerical values that measure similarities for all gene pairs. This output allows users that are interested in gene behavior to directly compare the function of any set of genes through distinct visuals in which we have shown throughout the manuscript. Further, the outputs of ALAN are gene lists, in which we can further study through GSEA¹¹. As one possibility, once genes are clustered using clustering approaches, ALAN can be used in parallel to visualize or compare why a series of genes fall into the same or different clusters.

As for benchmarking the two methods, it is unclear how to measure which method is superior in selecting all the genes that have similar functions. Both methods are computational tools that assess gene behavior based on

abundance level data. To biologically benchmark the outputs requires extensive functional knowledge of all the genes in a cluster group or ALAN ecosystem. To this degree, it is a challenge to benchmark which tool is more robust. Especially with the knowledge that gene functions may change in cancer cells.

With greater detail, here are some differences between ALAN and clustering tools we have discussed in the revised manuscript.

1. **Differences in output format.** Outputs of clustering approaches are cluster labels for each gene. As discussed in **Reviewer 1 Comment 1**, ALAN output matrices are derived by serial correlations. ALAN outputs contain a numerical value for every pair of genes ranging from -1 to 1, representing the degree of interaction between all gene pairs.
2. **Measuring gene similarity.** Hierarchical clustering approaches measure gene similarity by computing a distance between two genes. This measure of distance depends only on the expression values of the gene pair in question. On the other hand, ALAN measures the similarity between the behavior of two genes by computing the strength of correlation between the pair of genes as based on their expression patterns as compared to all other detectable genes (discussed in **Reviewer 1 Comment 1**)
3. **Mechanisms of gene grouping.** Hierarchical clustering approaches group genes into clusters based on a measure of distance between two genes. The number of groups can be user defined. ALAN outputs do not inherently identify clusters of genes, but instead return measures of similarity in the behavior of gene pairs. In our study, we identified groups of genes with similar behavior from an ALAN output by identifying a gene of interest and querying the ALAN output for those genes that share an ALAN score above a set threshold (0.7 was used in the manuscript). In this study, we also considered biological functions when setting this threshold.
4. **True biological validation of interactions.** Importantly, while methods of clustering and ALAN both measure gene similarity, strategies beyond computational approaches are required to truly “validate” each of these computational approaches to identify biological interactions. This includes assays that test protein interactions and functions (ChIP-seq, gene perturbation, or functional assays). For these reasons, throughout our study, we highlighted ALAN gene interactions that have already been established by multiple laboratory approaches.

To reflect these changes, in the revised manuscript, we have made the following amendments to the text:

Introduction section (page 2, lines 75-77), “Hierarchical clustering approaches group genes based on similarities. These outputs establish distances between two genes based on the expression values of the gene pair in relation to all other genes.”

Discussion section (page 8, lines 383-390), “Hierarchical clustering methods like ward.D2⁵⁰ and hierarchical tree cutting tools such as cutreeDynamic⁵¹ use metrics of gene similarity to assign genes into distinct groups. The expression patterns of all genes contribute to the clustering and the outputs of clustering approaches are cluster labels for each element. ALAN outputs do not inherently identify clusters of genes, but instead measure of similarity of all possible gene pairs based on numerical values. This provides an advantage to users that wish to visualize the relationship between a few specific genes. Users should consider the distinct utility of these parallel approaches as these functions may complement one another. As one example, a user may utilize ALAN to depict why certain genes are in the same or separate gene clusters.”

Discussion section (page 8, lines 402-404), “Regardless, while numerous approaches can be used to compare gene behavior, strategies including laboratory-based approaches are ultimately required to validate true biological functions.”

Reviewer #2 (Remarks to the Author):

Bergom et al., present here an Algorithm for Linking Activity Networks (ALAN) computational pipeline for a better understanding of context-specific cancer therapy. Applying three cancer types, mainly focusing on prostate cancer were used to present to show the application of the ALAN computational pipeline. The authors were able

to recapitulate several known associations or biomarkers in prostate or breast cancer. While their questions are interesting, there are several concerns that should be addressed prior to publication:

Introduction.

1. The authors pointed out the current limitations of ML/AI-based algorithms, especially focusing on PNET. On page 2: "We previously applied PNET, a neural-networking algorithm, to identify somatic driver events using DNA-sequencing data from primary prostate cancer and mCRPC. We identified one genomic feature, MDM4 amplifications, that represented a promising precision therapy target in mCRPC. The architecture of PNET was based on 3,007 pre-curated biological pathways that were not cancer-type specific. Overall, while these AI/ML approaches predict risk of occurrence from tumor samples, they include user/machine bias. This is because prior to analyses, users must pre-define the selectable features in order to build the model's architecture. The selectable features are often based on known gene signatures and static definitions of biological pathways. For these reasons, the interoperability and outcomes are highly contingent on pathway structures, which will be inaccurate when contexts change. Overall, an informatic driven solution that considers the context specificity could enhance the learning capacity of AI/ML tools." This part needs to improve for a more logical introduction.

We thank **Reviewer 2** for this comment.

The **Introduction** has been revised to logically introduce ALAN as an algorithm to understand gene behavior.

Based on this comment and **Reviewer 1 Comment 1** and **Comment 4**, we have revised the **Introduction** to discuss several computational tools that are currently used to compare gene behavior.

To note, we are classifying ALAN as an alternative to AI/ML tools that can be used to enhance the applications of AI/ML.

As part of the revised manuscript, we have conducted the following:

Introduction section (page 2, lines 63-77), "Many computational approaches have been developed to understand and interrogate gene behavior based on abundance level data from transcriptomic or proteomic approaches. Correlation approaches identify similar patterns of gene expression across patients between gene pairs. This method does not consider expression values outside of the gene pair of interest. Hierarchical clustering approaches group genes based on similarities. These outputs establish distances between two genes based on the expression values of the gene pair in relation to all other genes. Enrichment approaches such as Gene Set Enrichment Analysis (GSEA)²⁰, identify enrichment of functionally related gene sets when comparing two biological states. This approach relies on static definitions of biological pathways. In certain cases, there are true differences in a signaling pathway when a gene is active in different tissues or disease states, and this activity would be better measured through non-conventional or modified gene sets. Lastly, artificial intelligence and machine learning (AI/ML) are used to study gene behavior in many ways. These algorithms often stratify genes into classes²¹⁻²³ in which genes that fall into each class behave similarly. In the case of AI/ML, the grouping of the genes may not be transparent to a biologist, and the outcomes are influenced by the user-defined architecture of the specific model. Altogether, an approach that considers the global gene expression patterns as well as context cues could work with all such tools and enhance understanding of gene behavior through genomic datasets."

The authors need to clarify these points:

2. Are there any other ML/AI-based algorithms beyond PNET? Can we generalize the limitations only using the PNET example?

To the reviewer's question, there are many AI/ML algorithms.

We have cited 2 additional algorithms in the **Introduction** and **Discussion** sections that are utilized in studying expression data. We have discussed that there are certain features of AI/ML tools in which ALAN can enhance in each of these instances. We have replaced the original AI/ML citations that were included, since they were not utilized to study gene expression.

In P-NET we constructed hidden layers we previously termed “biologically informed layer” mapped based on curated pathways¹⁵. This assumes that pathways act the same across tissue type and cancer stage. We previously used P-NET as a classifier to identify genes relevant to prostate cancer metastasis. Our work with ALAN highlights the ability of ALAN to identify and compare prostate tissue type or prostate cancer stage specific behavior of genes, which can be rationally applied towards reconstructing the biologically informed layer of P-NET in future studies.

For other AI/ML tools in which we cited¹⁵⁻¹⁷, the study objectives were to classify genes based on specific features. Here, ALAN provides a utility towards understanding why certain genes are grouped together or separately.

In the **Discussion** section (page 8, lines 392-402),

“AI/ML tools are now available to study gene behavior with respect to outcomes. These tools classify genes based on specific features²¹⁻²³. Here, ALAN outputs can be applied on the same datasets to determine if genes that fall in the same class do in fact exhibit similar gene behavior. In our prior study, we developed P-NET to leverage cancer genomic data in order to predict critical “classifiers” that are potential biomarkers or even precision targets in metastatic prostate cancers (citation). The architecture of this neural-networking consisted of a multi-layered hierarchical network structure of genes where their interaction was inferred based on 3007 curated biological pathways. Given that we were studying distinct states of prostate cancer (primary and metastatic), it is possible that these preconceived pathways do not reflect the true relationship of genes within primary and metastatic prostate cancers. In cases like P-NET, we can utilize ALAN to construct the hidden layers. While we propose that the use of ALAN may enhance AI/ML tools in studying gene behavior, it remains to be determined how these approaches can be integrated and how this will impact the results.”

3. It seems like even PTEN included many different biological features such as copy-number changes, gene expression level, and mutations (from Fig1 in the original paper). What is the difference or strong point of ALAN method compared to PNET exactly?

We thank **Reviewer 2** for this comment.

We have clarified in **Reviewer 2 Comment 1** and **Comment 2** that while ALAN is not an AI/ML tool, it can act as a method to improve the biologically informed layers within P-NET.

Based on the interests in P-NET, we would like to highlight that despite the mention of gene expression levels in Figure 1 from the original paper, we previously only analyzed copy-number changes and gene mutations based on data. Specifically, the input data from the P-NET study was derived from Armenia et al¹⁸, which included only copy number and mutation status. At this time, we still do not have matching RNA data from many of these patients.

4. on page2, "where each cell contains a measurement of any molecular abundance such as gene expression, counts, or protein abundance"

It is unclear whether ALAN is allowed to use any kind of expression (or abundance data) either gene expression or protein abundance. If yes, could the authors also validate or provide the possibility to use it in the protein abundance data?

We thank **Reviewer 2** for this comment.

ALAN is indeed powered to use any kind of expression data or abundance data which includes mRNA expression as well as protein abundance. As part of the revised manuscript, we have demonstrated that even in protein data, ALAN is able to identify relationships between genes beyond standard gene correlation approaches.

We have discussed that AR and FOXA1 are in the same ALAN ecosystem (**Figure 4A**, also discussed in **Reviewer 1 Comment 1**). We have cited studies that AR and FOXA1 are proteins that bind to one another, interact with the same transcription regulatory elements, and regulate the Androgen Receptor signaling in prostate cancer^{2,3}. As part of the revisions, we identified a unique published proteomics dataset from Butrose et al¹⁹. Of this data, we aimed to perform a pairwise gene correlation analysis of protein expression levels between AR and FOXA1 protein levels (**Supplemental Figure 1A**). We also illustrated the results from ALAN analysis, which now accounts for the AR and FOXA1 interaction with all other proteins (**Supplemental Figure 1B**). As shown, the ALAN outputs were higher as compared to Spearman's or Pearson's correlation coefficients.

As part of the revised manuscript, we have made the following changes:

Results section (page 5, lines 218-223), "To support these results, we have also examined the AR and FOXA1 interactions through protein abundance data derived from prostate cancer biopsies²⁴. As compared to a pairwise Spearman correlation of AR and FOXA1 protein expression levels (**Supplemental Figure 1A**), ALAN identified that AR and FOXA1 are in a same ALAN ecosystem (**Supplemental Figure 1B**). This indicated that ALAN can be utilized to compare gene behavior through protein data as well as transcriptomic data."

Supplemental Figure 1A and B.

Supplemental Figures and Legends section (page 16, lines 538-542), "**Supplemental Figure 1.** The correlation between AR and FOXA1 is depicted from red (-1) to blue (+1) in primary prostate cancer protein abundance data. **A.** Input matrix protein abundance data was used where each dot is a patient and its x- and y- coordinates represent the gene expression of AR or FOXA1 respectively and the Spearman's correlation was calculated. **B.** ALAN Export Matrix 1 data was used where each dot represents an individual gene and its x- and y- coordinates represent the ALAN profile correlations with either AR or FOXA1."

5. on page2, "Entirely dependent on the input data, ALAN acts as an informatic-based assay that measures contextual gene behavior."

It seems like this method is mainly used gene expression data. It is not so clear how a single feature (gene expression) could perfectly measure context-specific behaviors. If this is true, comparisons between different cancer types will be necessary to prove it.

We thank **Reviewer 2** for this comment.

We recognize that the mention of “informatic-based assay” can lead to confusion, especially when considering the other terminologies that we have defined in this manuscript. We have removed this term in the revised manuscript.

As discussed in **Point 4** of **Reviewer 1 Comment 4**, we recognize that no informatic or computational approach will be “perfect”, as orthogonal laboratory approaches or clinical approaches are required for true functional comparisons.

With regard to context specific behavior, we highlight the ALAN outputs of MYC in our revised manuscript. In **Figure 3C** and **D** of the revised manuscript, we now compared the context specific gene behavior of MYC across multiple cancer types including ovarian, melanoma, lung, and pancreatic. Across these four cancer types, only 3.35% of genes were shared by another cancer type (**Figure 3C**). However, when applying a gene set enrichment approach (GSEA)¹¹, we found that the MYC ALAN network was enriched of MYC signaling in 4 out of 5 cancer types and all 5 subtypes of breast (**Figure 3E**). Altogether, ALAN identified highly distinct MYC profiles across these cancer types and breast cancer subtypes that were each enriched of Hallmark MYC activity. To this degree, ALAN identifies nuance gene relationships that are not detectable through conventional gene set enrichment approaches.

To reflect these points, we have included the following in the revised manuscript:

Results section (pages 5 and 6, lines 255 - 262), “While our initial observations examined the behavior of proto-oncogene *MYC* across various stages of prostate cancer, we sought to examine the behavior of *MYC* in other cancer types including melanoma, lung, ovarian, pancreatic, and breast (citations). To investigate whether *MYC* exhibited distinct behavior in these additional cancer types, we utilized ALAN to examine the *MYC* network signature in each context. Transcription data was obtained from The Cancer Genome Atlas (TCGA) on samples of melanoma, lung, ovarian, and pancreatic cancer and the five PAM50 molecular subtypes of breast cancer (Basal, ER2, Luminal A, Luminal B, normal-like). Each signature was built using the top 500 genes associated with *MYC* in that cancer type or cancer subtype. Interestingly, across the four cancer types, only 3.35% of genes are shared by at least one other cancer type (**Figure 3C**).”

Results section (page 6, line 266-273), “Given the distinct ALAN outputs of the *MYC* ALAN network signatures across multiple cancers and within cancer subtypes, we used GSEA to determine if the ALAN network signature was enriched of MYC activity. Despite having remarkably distinct ALAN networks, the MYC_UP.V1_UP gene signature shows high enrichment with all cancers and all 5 subtypes of breast cancer, except pancreatic (**Figure 3E**). These results indicate that ALAN is able to disambiguate the genes that exhibit similar behavior as MYC across multiple cancers and within subtypes and these ALAN gene networks were enriched of Hallmark MYC activity. This highlights the utility of ALAN in finding nuance differences of an oncogene when it is active across different cancer types and cancer subtypes”

Visual updates to include **Figure 3C**.

Figures and Legends section (page 13, lines 502-504), “**Figure 3.C.** The overlap between *MYC* ALAN network signatures across four distinct cancer types in TCGA (Melanoma, Lung, Pancreatic and Ovarian) are visualized using a Venn diagram.”

Updated **Figure 3E.**

Figures and Legends section (page 13, lines 505-508), “**Figure 3.E.** GSEA enrichment analysis with NES and FDR statistics of the MYC_UP.V1_UP gene set and the MYC ALAN Network from four distinct cancer types (Melanoma, Lung, Pancreatic and Ovarian) and five subtypes of breast cancer (Basal, Her2, LumA, LumB, Normal-like) from TCGA.”

6. on page3, "The ALAN pipeline next serially conducts two statistical correlation operations in which all intermediate matrices are exported for quantitative comparisons and visual assessment. A rank-based Spearman's correlation is then performed between every pair of columns (molecular IDs) which generates the first correlation matrix, Matrix 1. These relational profiles are further compared while building the second correlation matrix by performing a Pearson's correlation between every pair of columns (relational profiles) of Matrix 1 to generate Matrix 2. This subsequent correlation matrix provides a comparative functional network profile between molecular variables defined as the gene ecosystem of the input data." It is not so clear the purpose of the first and second steps and what's the difference between the two steps. Also, Figure 1 is not clearly present the difference between the two steps.

We thank **Reviewer 2** for this comment.

This comment was shared by **Reviewer 1 in Comment 1.**

In response to this, we have demonstrated the difference between the first and second step in **Reviewer 1 Comment 1**. We have also included visual updates to **Figure 1** to demonstrate these nuances and clearly describe the differences between the steps. For details, please refer to the discussion in **Reviewer 1 Comment 1**.

7. on page3, "ALAN pipeline without any pre-processing or normalization." The authors need to clarify why they do not need any pre-processing or normalization steps? Or how do they handle removing technical artifacts from different sources? If input samples include technical artifacts, are there any filtering steps to resolve them? It is also not clear which TCGA samples they used here since TCGA provided gene expression data either raw data and processed or normalized data.

We thank **Reviewer 2** for these comments.

The statement was based on our workflow in the original manuscript, in which we did not perform additional pre-processing on the datasets examined. We have revised this statement in the updated manuscript to minimize the confusion that ALAN may correct for necessary pre-processing steps.

As reviewers discuss, sequencing pipelines include pre-processing steps to minimize or eliminate technical artifacts. Since the inputs of ALAN are downstream of such pre-processing steps, technical artifacts based on upstream approaches are unlikely to be corrected. To this degree, ALAN still relies on the upstream pre-processing tools that may correct for such artifacts.

We have included this critical point in the following sections:

Materials and Methods section (page 2, line 98), "The input data must be in a format that allows for direct comparison between samples."

Discussion section (page 7, lines 354-359), "While we have demonstrated that RNA and protein abundance level data can be utilized as ALAN input matrices, upstream pre-processing steps are relevant in minimizing technical artifacts. To this degree, users must still incorporate such tools towards generating the ALAN input matrix prior to implementation. Due to the rank-based conversion in its initial analyses, ALAN is agnostic to the shape of the data's distribution. If the two assumptions of the input matrix are met, then no further data transformation is required to run the ALAN algorithm (e.g., log transformation or normalization)."

We have also included specifications as to which TCGA samples were analyzed:

Materials and Methods section (page 3, lines 111-112), "We included mapped expression profiles of the following studies as published on cBioPortal (log transformed and z-score normalized)"

Cancer type selection.

8. This work is mainly applied to prostate cancer and also two cancer types (breast cancer and ovarian cancer). Since the method is basically designed to use gene expression data, it is not clear why the authors applied their method to three specific cancer types. Especially, TCGA is composed of more than 30 cancer types, this method would be expected to use in a general manner. Could you clarify the definition of selected cancer types or apply this method to other cancer types?

We thank **Reviewer 2** for this comment.

We examined prostate, breast, and ovarian cancer in this study because of preexisting knowledge in these cancers that harbor aberrant steroid hormone signaling, such as AR. As noted throughout the manuscript, we have utilized existing knowledge of gene function to benchmark the biological findings (also discussed in **Reviewer 1 Comment 1**).

To address the reviewer's comment in studying other cancers, we have since examined additional cancer types including melanoma, lung, and pancreatic in **Reviewer 2 Comment 5** and discussed the context specific behavior of MYC, an important oncogene, in these settings as well as within different subtypes of the same cancer.

In summary, changes have been made in the text of the revised manuscript including **Results** section (pages 5 and 6, lines 255 - 262) and **Results** section (page 6, line 266-273). Visual changes have also been added including **Figure 3C and E**. For details, please refer to the discussion in **Reviewer 2 Comment 5**.

9. Also, it is not clear why the method description and analysis are differently applied to prostate cancer and breast or ovarian cancer. In the method section: GISTIC 2.0 calling of the AR focal amplicon (on page 4): only for prostate cancer Transcription data from normal prostate tissue samples (n = 245) was obtained from the GTEXportal (on page 3): only for prostate cancer

To clarify, GISTIC 2.0 is not a part of the ALAN algorithm. We specifically utilized GISTIC 2.0 to nominate genes co-amplified with AR, even in early stages of prostate cancer (**Figure 2A**). The GISTIC 2.0 outputs identified 9 genes focally amplified in Xq12. The purpose of identifying these genes was to examine how their behavior compares to AR as compared to other genes with known protein interactions such as FOXA1 and HOXB13^{2,3} (**Figure 2B**). FOXA1 and HOXB13 reside in 14q.21.1 and 17q21.32. The result in **Figure 2B** illustrated that despite being co-amplified, ALAN indicated that AR co-factors that reside on different chromosomes behaved more similarly to AR as compared to genes in the focal amplification.

After establishing this logic in **Figure 2**, we then utilized the stage specific prostate tissue data to examine context specific relationships between AR and MYC, which reside on distinct chromosomes (Xq12 and 8q24).

To reflect these revision points and points in **Reviewer 2 Comment 20**, we have included the following updates to the text:

Results section (page 4, lines 193-199), "We first tested the functionality of ALAN outputs by examining if AR would expectedly exhibit similar behavior as known co-factors. To benchmark the similarity of AR and its co-factors, we also compared these ALAN outputs to genes that are recurrently co-amplified with AR, even in primary prostate cancers. To identify co-amplified genes, in a cohort of 492 primary prostate cancer tumors, we found that AR resided in a focal amplicon on Xq12 with 9 additional genes based on GISTIC 2.0 outputs^{26,27} (**Figure 2A**). In mCRPCs¹⁰, we examined the ALAN outputs to compare gene behavior of AR, its co-factors FOXA1, HOXB13, GRHL2, NCOA2, to 8 other genes in Xq12 that were detected in the mCRPC transcriptome (**Figure 2B**)."

10. Additionally, in the discussion section, "Since ALAN recognizes all intricate gene-to-gene interactions across multiple stages of a cancer, or even across distinct cell types, we demonstrate that ALAN avoids the conventional and static definition of gene pathways in cancer." The authors generalized the application of the method. However, it is not so sure about multiple stages of a cancer or even across distinct cell types.

We thank **Reviewer 2** for these comments and tempered the statement as follows:

"Since ALAN recognizes genes with similar behavior across multiple stages of a cancer, or even across distinct cell types, we demonstrate that ALAN can be utilized to identify relationships that exist outside of the static definition of gene pathways in cancer." (page 7, lines 346-348)

With regard to multiple stages of cancer, we have demonstrated that ALAN identifies relationships between genes across multiple stages of prostate cancer as discussed in **Reviewer 1 Comment 1** and in **Figure 3A**.

With regard to distinct cell types, we have also demonstrated in **Reviewer 2 Comment 5** that ALAN identified distinct gene behavior across multiple different cancers and within different subtypes.

For details, please refer to the discussion in **Reviewer 1, Comment 1** and **Reviewer 2 Comment 5**.

Figures.

11. Figure1: in the top panel, there are two boxes indicating sample collection steps experimentally. However, this method is designed to use publicly available transcriptomics data sets and sample collection steps were not done in this manuscript. It is better to delete it. in the bottom panel, data processing is in the middle but it makes a misunderstanding of the order of process. 'Data processing' step would be better to be the first. The 'connectivity' panel is difficult to understand. What is the color/thickness/dashed line indicated? Also, the gene network/ecosystem should be defined in the figure more clearly.

We agree with **Reviewer 2**. The revised **Figure 1** focuses on features that specifically describe ALAN. We have also added detailed text to explain the depicted elements in both the visuals in **Figure 1** and in the **Figure legend**.

Our updates include:

We have removed the depiction of the publicly available transcriptomic data sets and sample collection steps. We agree that the depiction originally included workflows that are not part of ALAN.

We have arranged the panels per request of the reviewer. What was originally the “Data Processing” column is indeed shown 1st. Because all prior panels have been removed, we no longer specifically highlight this section as ALAN “Data Processing”. These revisions can be found in the revised **Figure 1** under “**Applying Algorithm for Linking Activity Networks (ALAN)**”.

The “**Connectivity**” is now labeled “**Theoretical Connectivity of ALAN**” as this visual model depicts how ALAN constructs connections between genes. Specific to this comment, in the **Figure Legends** (page 10, lines 468-479), we state that, “Individual genes are depicted as circles with their respective names (A, B, C, and D). The strength of the correlation between two genes is represented by the thickness of the line. In ALAN Profiles, Profile A (blue) and Profile C (grey) depicted for genes A and C, where the strength of the correlation between the expression patterns of genes (A-A, A-B, A-C, A-D, or C-A, C-B, C-C, C-D) across all samples is represented by the thickness of the line. ALAN Profiles are derived from gene expression correlations, represented by solid lines, across all pairs of genes. In ALAN Network, Network A (blue) and Network C (grey) are depicted for genes A and C, where the strength of the correlation between ALAN profiles (A-A, A-B, A-C, A-D, or C-A, C-B, C-C, C-D) across all genes is represented by the thickness of the line. ALAN Networks are derived from ALAN Profile Correlations, represented by dashed lines, across all pairs of ALAN Profiles. In Depiction of Nominated ALAN Ecosystems, multiple ALAN networks are compared in a two-dimensional space where Networks A and C being more similar are closer together on the plot, whereas shaded dots that are not highlighted represent other ALAN networks for all detected genes.”

We have also made visual updates to “**Theoretical Connectivity of ALAN**” in **Figure 1** for clarity.

In addition to these updates to **Figure 1** and the **Figure Legend**, we have also updated our language in the revised manuscript. We have described ALAN gene profiles, ALAN gene networks, and ALAN gene ecosystems in the **Materials and Methods** (page 3, lines 127-129), **Materials and Methods** (page 3, lines 130-133), and **Materials and Methods** (page 3, lines 134-136).

Minor comments:

12. In several sentences, references were not correctly addressed. For example, - We 282 examined scRNA seq data obtained from paired biopsy samples from one patient pre-enzalutamide treatment and 283 post-therapy resistance (on page6: Authors need to clarify which data set are you referring to here).

We appreciate this comment by **Reviewer 2**.

In the revised manuscript, we have clarified that the patient samples used in this analysis were isolated from the single cell RNA sequencing dataset from citation number 25, He, M.X. et al.

13. Many studies have accrued -omic data from tumor-derived specimens, including DNA (genomics), RNA (transcriptomics), protein/phosphoprotein (proteomics/phosphoproteomics), epigenetics (epigenomics), or even metabolites (metabolomics) [on page2: Authors need to point out details examples]

We thank **Reviewer 2** for this comment.

In the revised manuscript, we have clarified the resources and included citations in which each type of omic data has been accrued. We have added the following statement to the **Introduction** section (pages 1 and 2, lines 49-54):

“Consortium efforts have accrued DNA (genomics) data based on efforts including The Cancer Genome Atlas (TCGA) and AACR GENIE⁹, RNA (transcriptomics) data have been accrued based on efforts including TCGA, Stand Up 2 Cancer (SU2C)¹⁰, and Genotype-Tissue Expression (GTEx) Project¹¹. Proteomic/phosphoprotein data can be found in data portals including the in silico human Surfaceome¹² and the Human Protein Atlas¹³. Lastly, epigenomic data and metabolomic databases include Cistrome-GO¹⁴ and the Consortium of Metabolomics Studies (COMETS)¹⁵.”

14. AI and ML also have high potential to enhance information obtained from molecular diagnostic 69 assays by deconvoluting relationships between patient outcomes, genes, and signaling pathways (on page2: need to address current articles based on AI/ML).

We thank **Reviewer 2** for this comment.

In the Revised manuscript, we now include a brief introduction of AI/ML tools in the **Introduction** section and indicate how ALAN can be used to potentially enhance AI/ML approaches in the **Discussion** section. This includes 2 additional citations.

As part of this revision, we have discussed how ALAN could enhance these additional AI/ML approaches. Please refer to **Reviewer 2 Comment 1** and **Reviewer 2 Comment 2**.

15. What is the correct name of PNET or P-NET? Because the name in the original paper is P-NET not PNET.

We thank **Reviewer 2** for recognizing this mistake.

The correct name is P-NET. We have since made amendments in the revised manuscript.

16. In this manuscript, some terminologies were commonly used without clear definitions such as ecosystem, gene network, gene interaction, or gene-gene interactions. It makes it difficult to understand the manuscript since the meaning is not perfectly matched with the common definition in the field. For example, gene interaction usually indicates genetic interaction that could be a co-occurring or mutually exclusive relationship. Therefore, the authors need to clarify them correctly. There are several examples that need to improve.

We thank **Reviewer 2** for this comment that has greatly improved our language in the manuscript.

In our revised manuscript, we now specify that our use of ecosystems and gene networks is specific to a form of ALAN output (ALAN ecosystem, ALAN gene network). Specifically, ALAN ecosystems and ALAN gene networks are further defined when we first mention them in the **Results** and in the **Materials and Methods** sections.

Because the term “interaction” is broadly used in multiple areas of biological research, we now omit the use of “gene interaction” throughout the revised manuscript.

We have also described in the first paragraph of the **Detailed Responses to Reviewers** specific definitions utilized in the manuscript including the ALAN algorithm, ALAN networks and ALAN ecosystems (First page in this document).

These specific definitions were added to the following sections in the manuscript:

Materials and Methods section (page 3, lines 127-129), **ALAN Profile**: An ALAN gene profile results from the ALAN Output – Matrix 1. For each gene, this represents the expression pattern of one gene compared to another gene with respect to all patients. Comparisons between ALAN profiles are used to generate ALAN networks in ALAN Output - Matrix 2.

Results section (page 4, lines 169-173), “The first step of the ALAN algorithm uses a ranked based association method to generate ALAN profiles which are contained within ALAN Output – Matrix 1. In this matrix, the correlation coefficient represents the similarity between gene expression patterns across all patients for two genes. The correlation coefficients for a single gene against every other gene represents an ALAN profile.”

Materials and Methods section (page 3, lines 130-133), *ALAN network*: An ALAN gene network results from the ALAN Output - Matrix 2. For each gene, this represents the behavior of one gene with respect to all other genes detected in the dataset. Throughout the manuscript, we have compared the behavior of two or more genes by directly comparing their ALAN gene networks.

Results section (page 4, lines 173-176), “The second step of the ALAN algorithm uses a linear based association method to generate ALAN gene networks which are contained within ALAN Output – Matrix 2. In this matrix, the correlation coefficient represents the similarity between the behavior of one gene with respect to all other genes detected in the dataset.”

Materials and Methods section (page 3, lines 134-136), *ALAN Ecosystem*: An ALAN ecosystem encompasses a set of genes with similar gene behavior. Based on the user’s genes of interest, ALAN ecosystems can be identified by isolating all pairs of these genes with an ALAN gene profile correlation above 0.7.”

Results section (page 4, lines 176-177), “Genes with highly similar networks as indicated by correlation score of above 0.7 share an ALAN gene ecosystem.”

17. On page 2, While current informatics tools “classify” signaling pathways and biomarkers, understanding of gene behavior is an entirely distinct scientific objective that requires a tool that evaluates biological activities of the same gene in multiple contexts. Gene networks represent a mechanism to measure how one gene behaves. -> There is no definition of the gene network. Also, the network usually indicates the overall behavior, not a single (one) gene.

As discussed in **Reviewer 2 Comment 16**, our reference of gene network is specific to ALAN outputs. We have further clarified these definitions in the context of ALAN.

Our definition of an “ALAN gene network” for a gene is indeed specific to how one gene behaves. However, ALAN derives this behavior based on its relationship with all other genes. For the specific definition of ALAN network, please refer to **Reviewer 2 Comment 16**.

In the case of our manuscript, genes with similar “ALAN gene networks” can in fact have similar overall behavior. Please consider our detailed response regarding the three types of gene relationships in **Reviewer 1 Comment 1**. Here we highlighted groups of genes with similar ALAN gene networks that exhibit similar overall behavior. They may be co-factors that bind each other, genes that regulate the same signaling pathway, or genes with the same resistant function.

18. On page2, We can represent gene networks as the summation of the varying degrees of all gene-to-gene interactions using RNA-sequencing data across many patient samples.

As based on **Reviewer 2 Comment 16** and **17**, our mention of a gene network is specific to ALAN outputs.

To reflect that this and the prior comments throughout this revision, we have amended the sentence for accuracy (page 2, lines 81-83):

“For a gene, the ALAN gene network encompasses information of the varying degrees of how a gene is related to all other genes based on expression data across many patient samples.”

19. On page7, allows direct comparisons of informatics-based phenotypes -> what are informatics-based phenotypes?

We thank **Reviewer 2** for this comment.

We have removed the use of “informatics-based phenotypes”. In also considering **Reviewer 2 Comment 5**, which discusses “informatic based assay”, we recognize that we must limit key terms since we have already defined several terms specific to ALAN outputs (**Reviewer 2 Comment 16, Comment 17**).

20. Several sentences are difficult to understand due to the inconsistent description. For example. On page 4, "We first tested the functionality of ALAN to interpret projected interactions between AR and known co-factors in mCRPC patients, of which many are intrachromosomal. To map co-amplified genes of AR, we utilized outputs from GISTIC 2 to identify the recurrent focally amplified region of AR in 492 primary prostate cancer tumors (Figure 2A). AR resides in a focal amplicon on Xq12 with 9 additional genes where only 8 of the 9 genes were detected in this cohort. We first examined if ALAN could predict known AR interactions and if genomic interactions, such as focal amplifications, were indicative of similar gene profiles and gene networks in mCRPC patients." Here, it is not so clear which data sets they used here between mCRPC patients and primary prostate cancer tumors.

We thank **Reviewer 2** for this comment.

We have revised the statement for clarity, to adhere to other revision comments, and to include citation of the datasets. Particularly, the rationale of the analyses and the revised text is detailed in **Reviewer 2 Comment 9**. To summarize, our intentions were to demonstrate that AR co-factors exhibited greater degrees of similarity as compared to genes that are recurrently co-amplified with AR even at early stages of prostate cancer.

1. Abida, W., *et al.* Genomic correlates of clinical outcome in advanced prostate cancer. *Proc Natl Acad Sci U S A* **116**, 11428-11436 (2019).
2. Pomerantz, M.M., *et al.* The androgen receptor cistrome is extensively reprogrammed in human prostate tumorigenesis. *Nat Genet* **47**, 1346-1351 (2015).
3. Pomerantz, M.M., *et al.* Prostate cancer reactivates developmental epigenomic programs during metastatic progression. *Nat Genet* **52**, 790-799 (2020).
4. Bai, S., *et al.* A positive role of c-Myc in regulating androgen receptor and its splice variants in prostate cancer. *Oncogene* **38**, 4977-4989 (2019).
5. Sharma, N.L., *et al.* The androgen receptor induces a distinct transcriptional program in castration-resistant prostate cancer in man. *Cancer Cell* **23**, 35-47 (2013).
6. Gao, L., *et al.* Androgen receptor promotes ligand-independent prostate cancer progression through c-Myc upregulation. *PLoS One* **8**, e63563 (2013).
7. Consortium, G.T. The Genotype-Tissue Expression (GTEx) project. *Nat Genet* **45**, 580-585 (2013).
8. Hwang, J.H., *et al.* CREB5 Promotes Resistance to Androgen-Receptor Antagonists and Androgen Deprivation in Prostate Cancer. *Cell Rep* **29**, 2355-2370 e2356 (2019).
9. Bluemn, E.G., *et al.* Androgen Receptor Pathway-Independent Prostate Cancer Is Sustained through FGF Signaling. *Cancer Cell* **32**, 474-489 e476 (2017).
10. Han, G.C., *et al.* Genomic Resistance Patterns to Second-Generation Androgen Blockade in Paired Tumor Biopsies of Metastatic Castration-Resistant Prostate Cancer. *JCO Precis Oncol* **1**(2017).
11. Subramanian, A., *et al.* Gene set enrichment analysis: a knowledge-based approach for interpreting genome-wide expression profiles. *Proc Natl Acad Sci U S A* **102**, 15545-15550 (2005).
12. Nelson, P.S., *et al.* The program of androgen-responsive genes in neoplastic prostate epithelium. *Proc Natl Acad Sci U S A* **99**, 11890-11895 (2002).
13. Murtagh, F. & Legendre, P. Ward's Hierarchical Agglomerative Clustering Method: Which Algorithms Implement Ward's Criterion? *Journal of Classification* **31**, 274-295 (2014).

14. Langfelder, P., Zhang, B. & Horvath, S. Defining clusters from a hierarchical cluster tree: the Dynamic Tree Cut package for R. *Bioinformatics* **24**, 719-720 (2008).
15. Elmarakeby, H.A., *et al.* Biologically informed deep neural network for prostate cancer discovery. *Nature* **598**, 348-352 (2021).
16. Chaudhary, K., Poirion, O.B., Lu, L. & Garmire, L.X. Deep Learning-Based Multi-Omics Integration Robustly Predicts Survival in Liver Cancer. *Clin Cancer Res* **24**, 1248-1259 (2018).
17. Cascianelli, S., Molineris, I., Isella, C., Masseroli, M. & Medico, E. Machine learning for RNA sequencing-based intrinsic subtyping of breast cancer. *Sci Rep* **10**, 14071 (2020).
18. Armenia, J., *et al.* The long tail of oncogenic drivers in prostate cancer. *Nat Genet* **50**, 645-651 (2018).
19. Sinha, A., *et al.* The Proteogenomic Landscape of Curable Prostate Cancer. *Cancer Cell* **35**, 414-427 e416 (2019).

REVIEWERS' COMMENTS:

Reviewer #1 (Remarks to the Author):

The authors have favorably addressed my comments.

Reviewer #2 (Remarks to the Author):

The revised version improved all the previous questions, concerns, and uncertainty very clearly. It has clearly solved their technical issues and the method section has been improved to support their algorithm in a correct way.